# Industry funding of patient organisations in the UK: a retrospective study of commercial determinants, funding concentration and disease prevalence

Arianna Gentilini  , Iva Parvanova

Department of Health Policy, The London School of Economics and Political Science, London, UK

**Correspondence to**
Arianna Gentilini;
a.gentilini@lse.ac.uk

## ABSTRACT

**Objectives** To assess the relationship between UK-based patient organisation funding and companies' commercial interests in rare and non-rare diseases in 2020.

**Design** Retrospective analysis of the value and volume of payments from pharmaceutical companies to patient organisations in the UK matched with data on the conditions supported by patient organisations and drugs in companies' approved portfolios and research and development pipelines.

**Setting** UK.

**Participants** 74 pharmaceutical companies making payments to 341 UK-based patient organisations.

**Main outcome measures** Alignment between the commercial interests of pharmaceutical companies and the disease area focus of patient organisations; difference in the volume and value of payments to patient organisations broken down by prevalence of conditions; industry funding concentration, measured as the number of companies funding each patient organisation, the share of overall industry funding coming from each contributing company and the share of industry funding of each organisation comprised by the single highest payments.

**Results** 1422 payments were made by 74 companies to 341 patient organisations. Almost all funds (90%) from pharmaceutical companies were directed to patient organisations that are aligned with companies' approved drug portfolios and research and development pipelines. Despite rare diseases affecting less than 5% of the UK population, more than 20% of all payments were directed to patient organisations which target such conditions. Patient organisations focusing on rare diseases relied on payments from fewer companies (p value=0.0031) compared to organisations focusing on non-rare diseases.

**Conclusions** Companies predominantly funded patient organisations operating in therapeutic areas relevant to companies' portfolio or drug development pipeline. Patient organisations focusing on rare diseases received more funding relative to the number of patients affected by these conditions and relied more heavily on payments from fewer companies compared to organisations targeting non-rare diseases. Increased independence of patient organisations could help avoid conflicts of interest.

### STRENGTHS AND LIMITATIONS OF THIS STUDY

⇒ We develop a methodology to determine the concordance between commercial interests of pharmaceutical companies and disease areas supported by patient organisations.

⇒ We present a comparative analysis of industry funding to patient organisations depending on the prevalence of the disease(s) they support.

⇒ Our analysis focuses on a recent time period which might differ from historical trends.

## INTRODUCTION

Patient organisations—not-for-profit organisations mainly composed of patients and/or caregivers that represent and support the needs of patients or caregivers[1,2]—play an important role in the development, regulatory review and adoption of new drugs.

During research and development, patient organisations effectively advocate for resources to be directed to conditions where unmet need is highest.[3,4] Patient organisations support research design and planning, helping to identify patient-relevant study endpoints.[4] Patient organisations also represent patient views and preferences at the time of regulatory review and health technology assessment of new drugs.[5,6] For example, during technology appraisals conducted by the National Institute for Health and Care Excellence (NICE), which makes funding recommendations for the English National Health Service, patients and organisations representing the interests of patients, provide testimonies of their first-hand experiences on how the disease affects them and those around them.[7] Finally, when drugs are launched, patient organisations contribute to dissemination of research results to patient

BMJ

community and clinicians, and offer support and information on therapies available.[4 8]

Given the increasingly important role of patient organisations it is vital to understand their financial ties with pharmaceutical companies. Previous studies documented the large number and high value of payments from pharmaceutical companies to patient organisations,[2 8–10] the uneven distribution between and within therapeutic areas,[2 10] and the concentration of payments coming from a small number of pharmaceutical firms across multiple jurisdictions.[2 8–16]

What remains unknown is the alignment between the commercial interests of pharmaceutical companies and UK patient organisations' activities. Prior research has demonstrated that industry tends to prioritise commercially attractive conditions, and there is evidence to suggest that the marketing of a drug for a particular disease is associated with increased industry funding to patient organisations operating in that area.[2 10] However, such studies have typically been conducted in different geographical settings and have focused solely on marketed drugs, rather than examining the entire research and development pipeline of pharmaceutical companies. This is especially important given the lengthy timeline for drugs to reach the market,[17] as failure to consider drugs currently undergoing clinical trials may result in an incomplete picture.

Another gap in the literature relates to the dynamics between the pharmaceutical industry and patient organisations supporting rare versus non-rare conditions. In the UK, diseases are defined rare if they affect up to 5 people in 10 000.[18 19] The low prevalence of rare diseases and their different aetiology, coupled with the lack of interest from policymakers and manufacturers, who often prioritise more profitable and prevalent diseases, has necessitated the formation of patient organisations to advocate for the needs of rare disease patients.[20 21] The National Organisation for Rare Disorders, serves as the umbrella organisation for rare disease patients in the USA and has been instrumental in lobbying for scientific support and economic incentives to stimulate innovation in rare diseases.[22] This advocacy ultimately led to the passing of the Orphan Drug Act in 1983 in the USA and the European Union Regulation on Orphan Medicinal Products in Europe in 2000.[18 23]

Moreover, the limited availability and complexity of medical knowledge regarding rare diseases have also fostered patients and families affected by these conditions to come together to provide each other with support and medical expertise.[20 24] Patient organisations, which are primarily composed of patients and their caregivers, are in a unique position to share first-hand experiences that can inform research and regulatory decisions.[25] While this is true also for non-rare conditions, patient organisations' input in regulatory and health technology appraisals is particularly important in the context of rare diseases due to scarce evidence. For example, the Scottish Medicines Consortium provides opportunities for patient groups

and clinicians to have a stronger voice in the decision-making process for drugs used to treat rare and end-of-life conditions.[26] Similarly, three members of patient organisations sit in the Committee for Orphan Medicinal Products within the European Medicines Agency (EMA), the body responsible for granting orphan designations to drugs. Patient organisation-led registries that collect real-world data on disease progression can de-risk drug development for rare diseases.[20] While observational studies are common in non-rare diseases, they usually do not require the support of patient organisations' networks as patients are easier to identify and recruit.[3]

Finally, there has been limited exploration of the concentration of industry funding for patient organisations. A recent study by Mulinari and colleagues examined the average number of pharmaceutical companies making payments to Danish patient organisations,[15] while only one study has investigated the share of industry funding and the top drug company donor's share in UK patient organisations' income.[11] However, no study has specifically focused on the number of companies funding UK patient organisations, nor have they explored whether organisations' industry funding differs based on disease rarity.

Our paper aims to contribute to and expand on existing literature by examining the concordance between the commercial interests of pharmaceutical companies and patient organisations' activities in the UK. Using publicly available data on 2020 payments, we analysed the volume, value of payments to patient organisations according to their disease area of interest, with the objective of examining whether there are differences in funding patterns between rare and non-rare diseases. Lastly, we examined the concentration of industry funding, namely how many companies funded each patient organisation and the extent to which organisations might have been reliant on funding from a single company. Based on the reviewed literature, we formulated the following hypotheses:

*Hypothesis 1*: Regarding the concordance between the commercial interests of pharmaceutical companies and patient organisations' activities, we expect no difference between rare and non-rare patient organisations, under the assumption that companies are unlikely to fund organisations out of altruistic motives.

*Hypothesis 2*: Furthermore, we hypothesise that patient organisations targeting rare diseases would receive less overall funding due to their low prevalence. However, the existing incentives, high costs and consequent profitability of some orphan-designated drugs might affect the proportion of funding directed towards these organisations.[27 28]

*Hypothesis 3*: Considering the limited availability of drugs for rare diseases from a handful of manufacturers, we expect organisations focusing on these conditions to rely on payments of higher value and from fewer companies compared to those targeting more prevalent conditions.

## METHODS

### Data on industry payments

Disclosure reports on pharmaceutical companies' websites were our primary data source on payments from the pharmaceutical industry to UK patient organisations in 2020.[29] Disclosing payments to patient organisations is a requirement of Clause 29 of the Association of British Pharmaceutical Industry (ABPI) Code of Practice.[30] Specifically, the ABPI requires companies to keep a public record of any payment made to patient organisations on their website for a minimum of 3 years following the payment.[30] Companies that sign up to abide by the ABPI Code accept the jurisdiction of the Prescription Medicines Code of Practice Authority (PMCPA, code regulator), which also affects non-ABPI members operating in the UK.[30] Companies may be sanctioned by the PMCPA if they do not disclose their payments.[30] In an effort to increase transparency, Disclosure UK, an industry-led platform showing payments from pharmaceutical companies to healthcare professionals and organisations, launched a gateway in 2020 that collects hyperlinks to companies' disclosures of payments to patient organisations.[31]

First, we screened the websites of all pharmaceutical companies abiding by the ABPI Code, aided by the Disclosure UK patient organisations gateway. We retrieved payments information from the companies' websites to ensure that all payments were captured. Second, in light of a recent study unveiling that payments to patient organisations were misreported in the Disclosure UK database of payments to healthcare organisations (HCOs),[16] we screened the 2020 Disclosure UK HCOs database for payments to patient organisations.

If payments were not disclosed in the company's website nor in the Disclosure UK HCOs database, we assumed that the company did not make any payments to patient organisations in 2020, as commonly assumed in the literature.[2]

One investigator (AG) extracted payment disclosures from the companies' websites. These comprised the name of the patient organisation, the year when the payment was made, the reason for the payment and its value in the currency reported by the disclosing company. The 2020 Disclosure UK HCOs database was also screened, and recipients were matched to standardised patient organisations names. To ensure the data's accuracy, the final database was scanned for duplicates, but no such instances were found. When reported in different currencies, such as United States dollars, Swiss franc, Swedish krona, Norwegian krone and Danish krone, the value of the payment was converted to Great British pounds (GBP), using the Office of National Statistics historical yearly conversion rates.[32 33] All payments are reported in 2020 GBP. Two in-kind payments with a monetary value of zero were excluded from the analysis. Further details on variables' cleaning and coding can be found in the online supplemental material.

### Data on patient organisations

We retrieved data on patient organisations from their websites. Details on the therapeutic area they advocated for—proxied by International Classification of Diseases V.11 (ICD-11) codes—and whether the condition(s) was rare or non-rare were also extracted. Conditions were considered rare if they appeared in the Orphanet database of rare diseases,[34] which is the platform and repository of data on rare diseases and orphan drugs. Patient organisations that did not match the European Federation of Pharmaceutical Industries and Associations (EFPIA) definition of what constitutes a patient organisation were excluded from the analysis. We chose the EFPIA's definition for the following reasons. First, this corresponds the definition used in the wider peer-reviewed literature.[2 35] Second, other commonly used definitions, such as the one from the EMA, refer to the structure of patient organisations' governing bodies, which have to consist of over 50% patients.[36] Considering the high number of patient organisations included in our analysis, this requirement was challenging—if not impossible—to verify. Second, EFPIA's definition indicates what the pharmaceutical industry considers to be a patient organisation. Therefore, it helped us minimise selection bias issues as it includes a wide range of organisations. We excluded 66 payments to patient organisations that did not match EFPIA's definition. Subgroup analyses on excluded organisations can be found in the Online supplemental material.

### Determining commercial interests

We assessed whether—and the extent to which—a pharmaceutical company holds an interest in the disease supported by a patient organisation. We adapted the definition of 'interest' provided by NICE.[37] An interest is when there is, or could be perceived to be, an opportunity for a pharmaceutical company to benefit in the disease area where the patient organisation operates. This could include cases where the pharmaceutical company has a drug developed or in development for a condition targeted by the patient organisation, or where a drug in the company's portfolio or pipeline is restricted to a specific population affected by the disease supported by the patient organisation. We define portfolio as a group of drugs that a pharmaceutical company has already developed, gained regulatory approval for and is actively marketing or selling. Conversely, pipeline refers to the collection of drug candidates being developed by a pharmaceutical company, at various stages of development, from preclinical research to clinical trials.

To establish whether an interest existed or not, we first classified the conditions targeted by patient organisations to ICD-11 codes using the online ICD-11 database.[38] ICD-11 codes are mutually exclusive, exhaustive and are arranged as a single hierarchical tree, from level one (most general eg, *neoplasms*) to five (most specific eg, *plasma cell myeloma*). This means that specific diseases are nested within broader classifications. Although some

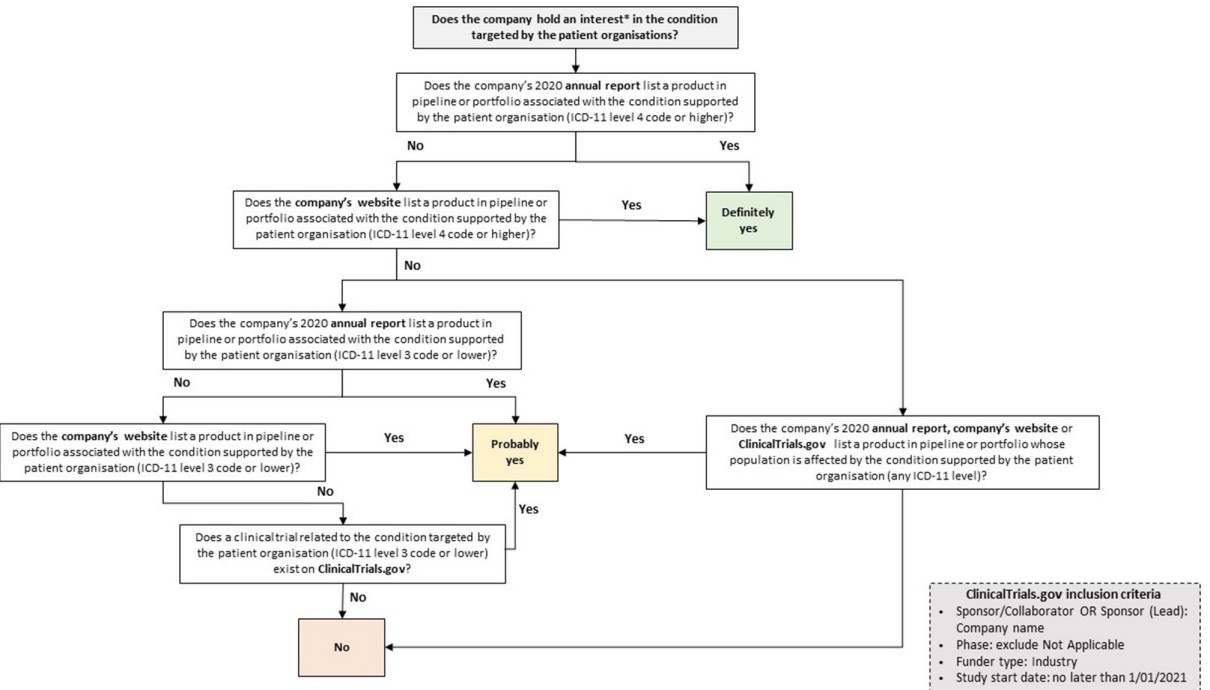

**Figure 1** Classification model to determine company interests in patient organisation funding. Note: An interest is when there is, or could be perceived to be, an opportunity for a pharmaceutical company to benefit in the disease area where the patient organisation operates. ICD-11, International Classification of Diseases V.11.

patient organisations, such as hospital charities, carers organisations and hospices, could not be matched to specific ICD-11 codes, they were included in the analysis to provide a comprehensive overview. As a result, the analysis presented results for both disease-specific and non-disease-specific organisations.

We then searched companies' annual reports, websites and the ClinicalTrials.gov registry to determine whether each company had an interest in the condition targeted by the patient organisation receiving the payment. Figure 1 schematically illustrates the approach taken to understand whether—and the degree to which—a company has an interest in the conditions (*definitely yes, probably yes, no*). For example, if *Company X* declares in its annual report having a drug in development for multiple myeloma and made a payment to *Blood Cancer UK*, this would be coded as *probably yes,* as the company has a product in its pipeline or portfolio nested within a broader class of conditions targeted by the patient organisation. Conversely, should *Company X* have made a payment to *Myeloma UK,* this would have been coded as *definitely yes,* as there is perfect alignment between the condition targeted by the patient organisation and by *Company X's* drug. Cases in which a company's interest in a certain condition could not be identified were coded as *no*. However, these instances might be due to limitations in data availability and therefore do not necessarily indicate that there was no company interest. Data on pharmaceutical companies' portfolio and pipeline were retrieved from their latest annual reports, company websites and ClinicalTrials.gov.[39]

One investigator (AG) initially coded all data, while the other (IP) blindly re-coded a 30% random sample of payments to validate the data collection process and minimise the risk of reporting errors. We followed this process when validating all data sources described above. Any disagreement was discussed until consensus was reached.

### Analysis of industry funding concentration

We assessed the concentration of industry funding received by patient organisations. In a prior study, Ozieranski and colleagues examined funding disparities among HCOs in the UK in 2015, using the Gini coefficient to assess the distribution of funding.[40] However, the authors acknowledged that the data preparation process presented challenges, limiting the analysis to payments from a single year. While this methodology has its advantages, we found that the time-consuming process of reshaping the data outweighed the benefits over using descriptive statistics. In particular, we calculated (1) the number of companies funding each patient organisation, (2) the share of overall industry funding to each patient organisation coming from each contributing company and (3) the share of industry funding of each organisation comprised by the single highest payment.

The Online supplemental fmateraial provides further details on the data collection and how the outcomes were constructed. Descriptive statistics and tests, such as ranges and Mann-Whitney U tests, were presented in the analysis. These statistics were preferred over the mean due to the skewed distribution of the data analysed. All analyses

and data visualisations were performed using Stata V.17 and RStudio (*ggplot2* package), respectively.

## Patient and public involvement

Patients were not involved in this study as our analyses focused on patient organisations as institutional actors rather than single patients with specific conditions. We plan to disseminate key findings from our analysis to patients and members of the public.

## RESULTS

In 2020, 74 companies made 1422 payments to 341 patient organisations, amounting to £22.6 million. Out of the total of 1422 payments made by pharmaceutical companies to patient organisations in 2020, 82% (1168 payments) with a value of £18 million were accurately disclosed on the companies' websites. The remaining 18% (254 payments) with a value of £4.6 million were reported in the Disclosure UK HCOs database. Among the companies, 24 out of 74 reported payments only on their websites, while 14 reported payments only in the Disclosure UK HCOs database and 36 reported payments in both.

Overall, *diseases of the nervous system* (£4.3 million) was the most funded therapeutic area over time, followed by *neoplasms* (£3.2 million) and *endocrine, nutritional or metabolic diseases* (£3.4 million). The conditions that received more funding in 2020 were multiple sclerosis (£1.7 million), followed by obesity (£1.4 million) and epilepsy (£1 million). Pfizer, Novo Nordisk, UCB, Novartis and Roche were the top five funders over the study period (figure 2). These companies contributed to more than one-third (36%) of all payments.

Table 1 summarises the number and value of payments to patient organisations.

## Companies' interest in payments to patient organisations

In 2020, 85% of all payments were directed to patient organisations that were judged to be aligned with their portfolio or pipeline. Only 15% of payments were made to organisations that focused on conditions that could not be linked to a product in the funder's portfolio or pipeline. Table 2 shows the volume and value of payments, broken down by the company's interest variable, overall and whether patient organisations targeted a rare or non-rare disease. Payments to patient organisations targeting a disease for which the company has a product developed or in development (*definitely yes*) made up 56% and 54% for patient organisations targeting rare and non-rare conditions, respectively. However, this difference was not statistically significant as anticipated in *Hypothesis 1* ($\chi^2 = 1.049$, p value=0.592).

The monetary value of payments coded as definitely yes accounted for 55% of the overall payment value. However, this was as high as 67% for patient organisations targeting rare diseases, versus 59% for organisations focusing on non-rare conditions. This difference was found to be statistically significant ($\chi^2 = 370.163$, p value=0.058). When payments coded as probably yes were included, the share increased to 90% and 97% for all patient organisations and disease-specific organisations only, respectively.

## Industry funding of patient organisations focusing on rare versus non-rare conditions

Of the £22.6 million in payments from industry to patient organisations, £4.6 million (21%; n=286) were directed to organisations focusing on rare diseases while £15.9 million (70%; n=952) to organisations supporting non-rare conditions. The remaining 9% was directed to non-disease-specific patient organisations, which were excluded from this analysis. Linking these results to *Hypothesis 2*, we observe that patient organisations supporting rare diseases received less but still substantial funding.

The most funded patient organisation overall in 2020 was the European Association for the Study of Obesity, receiving almost £1.5 million, followed by Epilepsy Society (£955 600) and Shift.MS (£588 451). Among the top 10 recipients overall in 2020, only one focused on rare diseases (Cystic Fibrosis Trust). However, it is worth noting that Blood Cancer UK, which focuses on malignant haematological malignancies including rare cancers, ranked seventh on the list.[41] The Cystic Fibrosis Trust (£445 229), The Society for Mucopolysaccharide Diseases (£358 037) and the International Patient Organisation for Primary Immunodeficiencies (£345 914) were the top three recipients focusing on rare diseases, followed by Myeloma UK with a slightly lower amount (£340 604).

Figure 3 shows therapeutic areas in order from most to least funded, broken down by rarity of disease targeted. In the case of organisations focusing on rare diseases, endocrine, nutritional or metabolic disease, neoplasms and diseases of the nervous system received most funds. Together, the top three most funded disease areas represented about half of overall funding (57%). When looking at the non-rare conditions that attracted most funding, multiple sclerosis was first (£1.7 million), followed by diabetes (£1.4 million) and epilepsy (£1 million). Cystic fibrosis, primary immunodeficiencies and lysosomal storage diseases, which include rare metabolic disorders such as Fabry and Gaucher diseases, received the highest funding overall, attracting £445 229, £363 998 and £358 037, respectively.

## Industry funding concentration

Each patient organisation received payments from a median of approximately one unique company, with 1 (IQR: 1–2) and 2 (IQR: 1–3) companies funding patient organisations targeting rare and non-rare diseases, respectively. However, this difference was not statistically significant (z=1.582, p value=0.114). Overall, the range of unique companies making payments to a unique patient organisation spanned from a minimum of 1 to a maximum of 13. The latter was recorded for Genetic Alliance UK, a national charity and an alliance of over 200

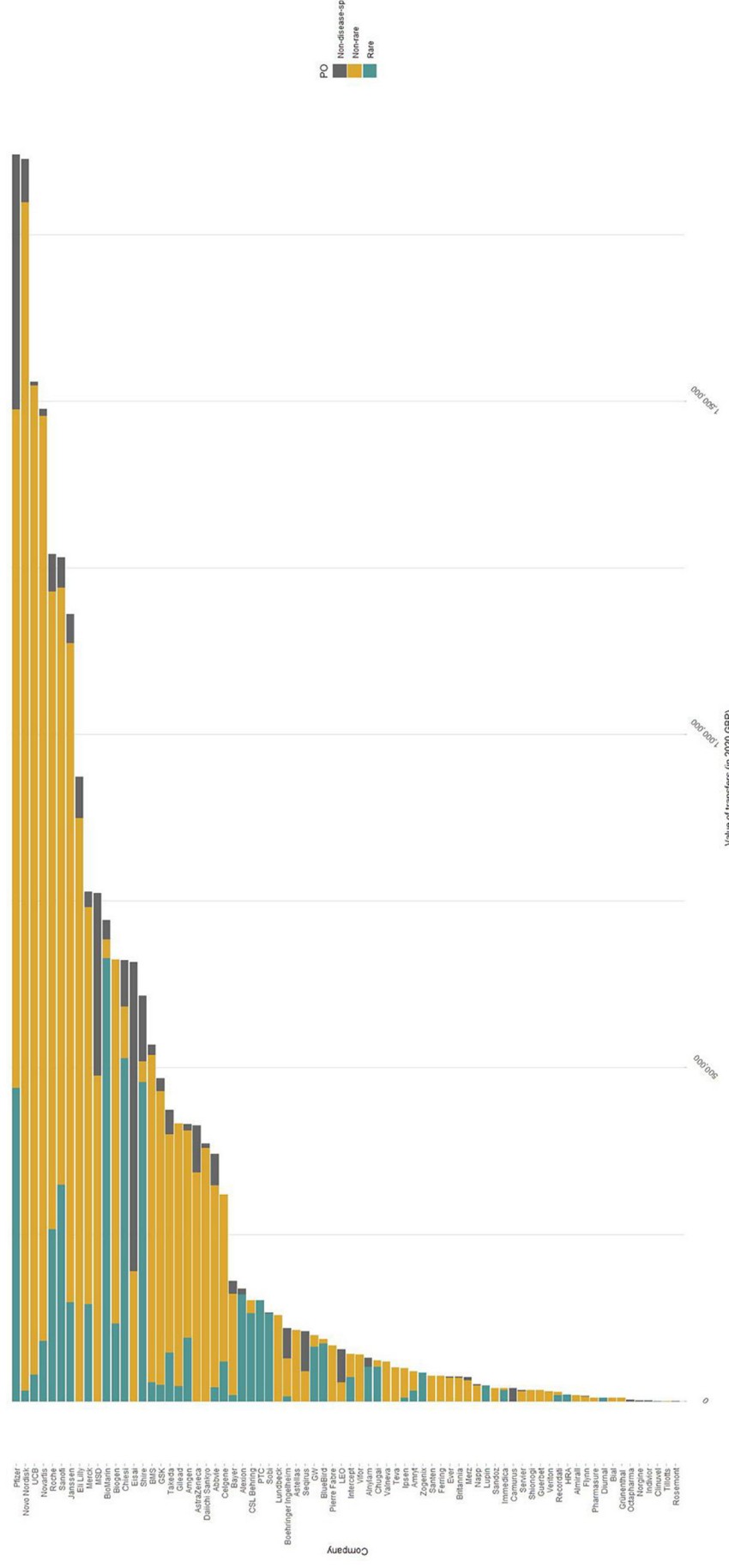

**Figure 2** Cumulative value of payments by receiving patient organisation type and funding company in 2020. Note: Non-disease-specific patient organisations include organisations that could not be matched to specific International Classification of Diseases V.11 codes or could not be classified as rare or non-rare, such as hospital charities, carers organisations and hospices. GBP, Great British pounds; PO, patient organisations.

Table 1  Number and value of payments from the pharmaceutical industry to UK patient organisations broken down by year and rarity of diseases

| Payment statistics | |
| --- | --- |
| Number of payments | 1422 |
| Median payment (IQR; overall) | £7943 (£1200–£15 000) |
| Median payment (IQR; rare) | £8775 (£2500–£15 965) |
| Median payment (IQR; non-rare) | £9060 (£1520–£16 850) |
| Value of payments (£; overall) | £22 577 314 |
| Value of payments (£; rare) | £4 629 779 |
| Value of payments (£; non-rare) | £15 875 662 |
| Number of pharmaceutical companies | 74 |
| Number of patient organisations | 341 |

Notes: All payments are expressed in 2020 Great British pounds. The supplemental materials detail the conversion rates used, which were retrieved from the Office of National Statistics website. Further details on how patient organisation data were cleaned and coded, please see the online supplemental materials . Please note that the number of pharmaceutical companies and patient organisations making and receiving payments across the study period refers to companies and organisations that made or received at least one payment, respectively.

Table 2  Volume and value of payments by company interests in 2020

| PO type | Company's interest | Volume; n (%) | Value: £ |
| --- | --- | --- | --- |
| Overall* | Definitely yes | 678 (48) | £12 529 514 (56%) |
| | Probably yes | 525 (37) | £7 700 069 (34%) |
| | No† | 219 (15) | £2 347 732 (10%) |
| Rare | Definitely yes | 161 (56) | £3 119 217 (67%) |
| | Probably yes | 115 (40) | £1 388 545 (30%) |
| | No† | 10 (4) | £122 017 (3%) |
| Non-rare | Definitely yes | 517 (54) | £9 410 297 (59%) |
| | Probably yes | 389 (41) | £6 056 915 (38%) |
| | No† | 46 (5) | £408 449 (3%) |

Notes: *Definitely yes* indicates payments directed to patient organisations that operated in a disease area (ICD-11 level 4 or higher) for which the company has a product in its portfolio or pipeline. *Probably yes* indicates directed to patient organisations that operated in a disease area (ICD-11 level 3 or lower) for which the company has a product in its portfolio or pipeline. *No* refers to directed to patient organisations that operated in a disease area for which no link could be found to the company's portfolio or pipeline.The higher the ICD-11, the more specific the condition. For example, if the ICD-11 level 4 is *plasma cell neoplasms,* level 2 would be *neoplasms of haematopoietic or lymphoid tissues.* Further details on how this variable was constructed can be found in the online supplemental material
*Please note that the *overall* results are not a sum of the *rare* and *non-rare* results, as they also include patient organisations that could not be classified in either group and are non-disease-specific.
†Please note that the *no* category of interest conservatively includes also interests that were considered as unclear.
ICD-11, International Classification of Diseases V.11 ; PO, patient organisations.

patient organisations, supporting those affected by rare genetic conditions.

In our sample, the median yearly payment of a company to a patient organisation comprised 24% of its overall industry payments (IQR: 9.5%–74%). When looking at patient organisations focusing on rare diseases, the median company contribution was as high as 30% (IQR: 11.6%–93%) versus 23% (IQR: 9.4%–65.8%) for non-rare conditions ($z=-2.164$, p value=0.031).

Finally, the share of industry funding comprised of the single highest payment per organisation amounted to an average of 67.5% (SD: 0.30) for all years, ranging from a minimum of 8.5% to a maximum of 100%. The highest value payment in the case of patient organisations targeting rare diseases made up a larger share of the overall industry funding (median: 71%, IQR: 43.5%–100%), despite not significant, compared with those focusing on more prevalent conditions (median: 62.5%, IQR: 34.7%–100%). While there was not a significant difference in the number of funding companies between patient organisations supporting rare and non-rare diseases ($z=-1.087$, p value=0.277) as stated in *Hypothesis 3*, the former relied on larger payments. Histograms illustrating the distribution of the statistics explored in this analysis can be found in the online supplemental materials.

## DISCUSSION

In this study, we evaluated the financial links between the pharmaceutical industry and patient organisations in the UK in 2020. This is the first study to document the almost-perfect concordance of pharmaceutical company interests and patient organisation funding in the UK. Almost all industry payments during our study period—in terms of both volume (85%) and value (90%)—were to patient organisations aligned with pharmaceutical companies' portfolios and pipelines. This share was even higher when considering only disease-specific patient organisations (97%). Despite rare diseases affecting less than 5% of the UK population, more than 20% of industry funding to patient organisations in 2020 was directed towards organisations focusing on such conditions (£4.6 million/£22.6 million). Finally, we found that patient organisations targeting rare diseases relied on payments from fewer companies but of higher value compared with organisations focusing on non-rare diseases.

The almost-perfect concordance between industry interests and patient organisation activities likely reflect the commercial attractiveness of conditions targeted by pharmaceutical companies.[2 42] Such close alignment between the interests of companies and patient organisations might undermine the credibility of patient organisations as perceived by the general public and might raise questions about patient organisations' inputs in regulatory

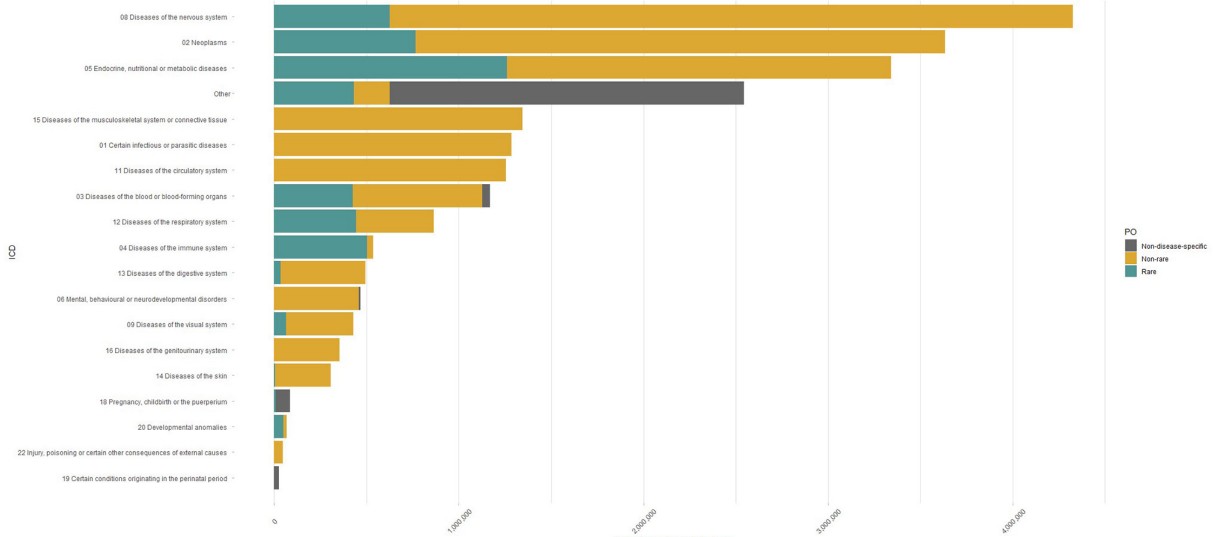

**Figure 3** Cumulative value of payments by patient organisation type and therapeutic area from in 2020. Note: Non-disease-specific patient organisations include organisations that could not be matched to specific International Classification of Diseases V.11 codes or could not be classified as rare or non-rare, such as hospital charities, carers organisations and hospices. GBP, Great British pounds.

and health technology appraisals.[9 43 44] Similarly, a study found that during NICE appraisal meetings fewer than 25% of all relevant financial ties between patient organisations and pharmaceutical companies were disclosed.[45] As discussed by the Mandeville and colleagues, this lack of transparency increases the risk of conflicts of interest not being properly detected and managed.

Our findings make an important contribution to the existing body of literature on industry funding of patient organisations. Ozieranski *et al* found that industry donated over £57 million to UK patient organisations from 2012 to 2016, an average of £11.5 million per year.[2] The authors also observed that payments were concentrated in commercially attractive therapeutic areas, with organisations focusing on cancer receiving more than 36% of overall payments.[2] However, the study did not examine whether companies were more likely to fund organisations that target diseases for which they have already developed or are currently developing products. Another earlier study examined payments to Swedish patient organisations and found an association between drug commercialisation and industry funding.[10] The authors did not take into account products in the companies' pipelines nor drugs that might have not yet launched in Sweden. Considering that patient organisations have an important role not only in the post-commercialisation phase but also in the R&D and approval stages. We therefore developed a replicable classification model to determine whether payments from companies were directed at organisations that were aligned with their portfolios and pipelines.

Patient organisations focusing on rare diseases can drive both supply of and demand for medicinal products due to their research, advocacy and education role.[4 46] As a result of their close ties with patients, these organisations have the credibility and power to educate patient communities, advocate for access to available therapies and raise awareness on the unmet need of certain conditions.[4 20 47] Although a large share of both the value and number of payments were directed to patient organisations focusing on rare diseases, most funds targeted commercially attractive rare conditions, such as multiple myeloma and cystic fibrosis, where the unmet need is relatively low compared with other rare conditions. These are diseases that have relatively high prevalence and for which 10 and 29 treatments, respectively, are currently approved for use in Europe.[34 48] Furthermore, rare diseases have proved a lucrative asset for pharmaceutical companies.[42] The additional market protection granted to orphan-designated product and the often higher willingness to pay from payers has led companies to increasingly focus on these medicines, which can offer a high return on investment.[27 28] This poses the risk of widening already existing health inequities, where severe and debilitating rare conditions that affect a small number of patients do not receive the resources they need and have to rely on limited public grants.[49]

Finally, our analysis showed that patient organisations focusing on rare diseases are funded by very few companies, relying on a single payment for over 70% of their industry-reported income. Despite the share of industry contributions among the overall patient organisation's income was found to be low in the literature,[11] this increases the risk of pursuing the company's commercial interests rather than objectively representing a patient body.[12] In this study we find that patient organisation received payments from a median of approximately one unique company (IQR: 1–3), ranging from 1 to a maximum of 13. This corresponds to an average of 2.6 (SD: 2.3) funding companies per patient organisation.

This is consistent with findings from a recent study investigating the distribution of payments from industry to Danish patient organisations, which found that on average, most organisations were funded by 2.6 (SD: 2.1) on average.[15]

These findings have important implications for policy and practice. To minimise conflicts of interests and maintain the integrity of patient organisations, particular attention should be paid to funding from companies in the period before or after a patient organisation has endorsed this company's product.[45] However, the duration of this period should be carefully evaluated to avoid overlooking more historical commercial ties.[50] One way of avoiding potential conflicts of interest is through increased transparency. Despite considerable progress on this front, especially in terms of reporting the monetary value of industry payments, there are still gaps in reporting.[51]

As highlighted in this and other studies, several companies misreport their payments to patient organisations.[16] Our study found that only 32% of companies disclose all of their payments correctly (ie, on their website), while the rest report them on both their websites and the Disclosure UK HCOs database (49%) or solely on the latter (19%). This duplication of reporting efforts makes it harder to achieve transparency and obtain a comprehensive overview of the financial relationships between companies and patient organisations. Therefore, efforts should be made to establish a unique repository for payments to patient organisations, similar to the one currently in place for physicians and HCOs.

Furthermore, the financial independence of patient organisations is fundamental to ensure that patients' interests are at the forefront of the organisations' agenda.[52] Compromising this independence can have a detrimental effect and distort public health priorities. For example, AbbVie-sponsored patient organisations were found to strongly oppose switching to biosimilars for HUMIRA, the company's blockbuster drug, in various countries.[15] Similarly, a recent investigation uncovered strong financial connections between Novo Nordisk and UK-based patient organisations that supported the approval of the company's latest obesity drug. This, alongside other ongoing investigations, culminated in the suspension of the company from ABPI.[53] The strong financial ties between Novo Nordisk and patient organisations, contributing to the NICE appraisal of the company's drug, raises serious concerns about these groups' independence and might ultimately harm patients.[50] Notably, our analysis found Novo Nordisk to be the second highest funder of patient organisations in term of value in 2020 for an amount of more than £1.8 million. In the long-term, policymakers should make sure that patient organisations receive adequate public funding regardless of whether they focus on conditions that are profitable for the industry. Such public funding is particularly important for patient organisations supporting rare diseases, as relatively few companies have financial links with patient organisations focusing on rare diseases, potentially creating high reliance on few high-value payments.

This study had limitations. First, the lack of mandatory reporting of payments to patient organisations by companies that do not comply with the ABPI Code is a major limitation of our analysis.[54] For example, our data set does not include payments by Vertex, a company with a rare-focused portfolio and a strong presence in cystic fibrosis.[55] Even for companies that are signatories of the ABPI Code, under-reporting of payments to patient organisations and removal of disclosure reports from the public domain has been observed.[13 56 57] Second, in our assessment of company interests, we made a conservative assumption that only patient organisations which target relatively narrow conditions were eligible to be coded as *definitely yes*. Despite this assumption, we concluded that more than half of payments were in therapeutic areas in which companies had a clear interest. Finally, our analysis focused on a recent though limited time period. While previous publications show similar trends in terms of the most funded diseases and absolute value of payments,[2 10] lending credibility to our analysis and underlying data, it is still unclear whether these trends hold over time and their generalisability to other periods.

There are several avenues which can be explored further to build on this analysis. While some of the previous literature on the topic has focused on the financial dependency of patient organisations' budgets from pharmaceutical funding,[11] whether this differs depending on the rarity of the disease targeted has not been explored. Due to the small number of patients affected by rare conditions, patient organisations that target such conditions may be less well-equipped to finance their activities via charitable events and may rely more heavily on contributions from pharmaceutical companies. Lastly, while our analysis did not evaluate the effect of COVID-19 on the financial dynamics between pharmaceutical companies and patient organisations, we expect that the pandemic had a substantial effect on the type, value and distribution of payments. Future research should examine the impact of COVID-19 on industry funding of patient organisations.

## CONCLUSIONS

Almost all industry funding of UK patient organisations in 2020 was in areas that were aligned with companies' approved drug portfolios and research and development pipelines. Pharmaceutical companies spent a larger amount on patient organisations focusing on rare diseases and these organisations relied on a small of companies for their funding.

**Acknowledgements** We thank Dr Huseyin Naci and Dr Olivier Wouters, both from the Department of Health Policy at the London School of Economics and Political Science, for providing comments on earlier drafts of this paper. We also acknowledge Dr Mylene Lagarde and Dr Panos Kanavos for the support provided when finalising the manuscript.

**Contributors** AG developed a preliminary version of the study and developed it further with IP. AG collected the data. AG and IP did the analysis, wrote and reviewed the manuscript. Both authors had full access to all of the data (including statistical reports and tables) in the study and can take responsibility for the integrity of the data and the accuracy of the data analysis. The corresponding author attests that all listed authors meet authorship criteria and that no others meeting the criteria have been omitted. AG is the guarantor.

**Funding** This research received no specific grant from any funding agency in the public, commercial or not-for-profit sectors. AG is funded by the ESRC for her doctoral research (Project reference: 2480077) and is employed by Imperial College London. IP is funded by the ESRC for her doctoral research (Project reference: 2311285) and is employed by Imperial College London.

**Competing interests** None declared.

**Patient and public involvement** Patients and/or the public were not involved in the design, or conduct, or reporting, or dissemination plans of this research.

**Patient consent for publication** Not applicable.

**Ethics approval** Not applicable.

**Provenance and peer review** Not commissioned; externally peer reviewed.

**Data availability statement** Data are available via the Dryad data repository at http://datadryad.org with the doi: 10.5061/dryad.fqz612jxd. Extra data are available upon reasonable request.

**ORCID iD**
Arianna Gentilini http://orcid.org/0000-0002-6943-6158

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
