## [Reviewer comments · BMJ Open]

ARTICLE DETAILS

TITLE (PROVISIONAL)	Industry funding of patient organisations in the United Kingdom: A retrospective study of commercial determinants, funding concentration and disease prevalence
AUTHORS	Gentilini, Arianna; Parvanova, Iva

VERSION 1 – REVIEW

REVIEWER	Mulinari, Shai Lunds Universitet, Sociology My partner is employed by ICON, a global Contract Research Organization whose customers include many pharmaceutical companies.
REVIEW RETURNED	31-Jan-2023

GENERAL COMMENTS	Thank you for the opportunity to review “Industry funding of patient organisations in the United Kingdom: A retrospective study of commercial determinants, funding concentration and disease prevalence” Comments in order of appearance in manuscript 1. p. 4. The authors claim patient organisations (POs) have roles “across all stages of drug development, approval and access”. However, it would be easy to find examples of stages where POs do not have roles, e.g., related to company decisions, marketing, and pricing. I suggest to reformulate this.2. p. 4. Some additional studies of industry funding of POs could be cited here to underscore the international nature of this literature, including from Canada, Australia, Finland and Denmark.3. p.4; lines 31-40. I believe the authors downplay some of the literature here. They claim that “whether companies fund patient organisations operating in therapeutic areas relevant to companies’ approved drug portfolios and research and development pipelines remains unanswered.” However, the Swedish study cited earlier in this paragraph looked at whether companies fund patient organisations operating in therapeutic areas relevant to the companies’ drug portfolios (see Table 6 in that paper). It is true that the study did not look at R&D pipelines, but it’s unclear if this makes any difference (see more below), and, moreover, the Swedish study looked at funding over a longer time-period (2014-2018), meaning some of the “research and development pipelines” had become part of “approved drug portfolios”.4. p. 4. The distinction rare/non-rare disease is central to this paper. However, the authors could do more to justify this through a more detailed review of the literature, and, based on this, develop
--

	some preliminary hypotheses or expectations. For example, there is some research suggesting that it might be more difficult for rare-disease POs to get non-industry funding (mentioned in the Discussion). Also, in Denmark, POs with fewer members (less than 100) relied more on industry funding, see 10.1016/j.healthpol.2022.11.003 5. p. 5; line 4-6. The authors might also want to check out the Danish study for this statement as it similarly addressed the question of concentration of funding. 6. p. 6; line 17-18. It is the PMCPA that sanctions companies, not the ABPI. The ABPI Board may suspend or expel a company but such cases are extremely rare. 7. p. 6: line 20-22. Why would companies report payments to UK POs in other currencies? 8. p. 7: line 17 & Fig. 1. a. Why did you restrict the study start date in Clinicaltrials.gov to 1/01/2021 when the funding you study took place between 2018-2020? Would it not have been more relevant to include trials running between 2018 and 2020, or even earlier, given the time it takes to develop and test a new drug? b. Related to the previous point: the authors should provide data on the results from their searches, e.g., add actual numbers in Figure 1. c. I think it would be helpful for others (and increase the relevance of the study) if the authors could also add a supplementary file with company interests across disease areas. d. It was unclear to me if the analysis also evaluated cases where companies did not fund POs in diseases linked to their commercial interests, as the Swedish study did, see comment above. If not, why? 9. p.9; line 7. The use of the word “significantly” might suggest to some that the change was statistically significant. 10. p.9; line 4. Most important comment: The substantial increase in funding in 2020 seems remarkable at first sight. I would have expected reduced funding of POs during Covid. However, what I think is going on is that the increase is an artifact of company reporting practices (or, alternatively, of the authors’ search strategy). Indeed, the number of reporting companies in 2018 is about 30 and in 2020 it is about 60 (see Supp Fig.1)! This strongly suggests that the study has major missing data problem which challenges the validity of the results. It is very surprising that the authors choose not to foreground and comment on this (here as well as in the discussion) and interpret all their calculations and findings in light of this very likely bias. E.g., it does not seem credible that AbbVie made no payments in 2018! Also, because non-reporting seems more common among smaller companies, which might be more likely to focus on rare diseases, it probably undermines the comparison between rare/non-rare, in addition to calculations of funding concentration. There are a number of studies that have looked at company reporting practices, including in the UK, as well as a recent Nordic comparative study which suggested that countries without centralized databases had higher levels of under- and non-reporting. The authors should re-evaluate their study in light of this missing data problem. 11. p.9; lines 5-7. Could you provide more information on specific diseases? That is, provide a breakdown of the broad disease areas (level 1) into their lower-level diseases (level 2 or below). This might allow you to make more granular interpretations of funding patterns and compare with other studied countries. Some
--	--

	information is provided later on p. 11 which could be moved here and expanded. 12. p. 9; line 11. I could not identify Supp Table 1 for this. Do you mean Supp Table 2? 13. p.9; lines 14-15. What would the results have been if you would only consider the “portfolio”? In other words, what difference does the “pipeline” make? And how much difference did the inclusion of trials make? This is important because you later claim that an advantage of this study is that it included the “pipeline”. See also comment above. 14. Table 1: consider not breaking down results per year. There are no obvious differences between years (other than the total sums). 15. Fig 3. “Figure 3 shows therapeutic areas in order from most to least funded”. This does not seem to be the order in the figure. 16. Fig 3. is slightly confusing because the order or rare/non-rare is different for different therapeutic areas (also Fig S2). 17. p. 11: lines 6-7: “From 2018 to 2020, the transfer to patient organisations targeting rare diseases increased more compared to those focusing on more prevalent conditions (80% vs 57%).” Could this be explained in part by less underreporting by smaller companies targeting rare diseases? 18. p.11; lines 16-19. See comment 11. 19. p.11; lines 16-19. The reference to Table 2 does not seem right here. Table 2 shows specific POs not disease areas. 20. Please re-check Table 2. I was surprised that single transfers would be this dominating. For example, according to Table 3, Chiesi made one and only one transfer to the CF Trust worth 1.3 million. However, I quickly checked Chiesi’s reports and the company had 3 payments in 2019. Another example: Epilepsy Society reportedly had two industry funders but Table 3 also reports that one payment accounted for 100% of the funding, which must be inaccurate. 21. p. 13: lines 6-10. Consider having the IQR values as %. 22. p. 13: lines 11-17. Why did you choose to calculate/report this per year rather than (also) across all 3 years? In general, it is unclear when and why you choose to report data across all years and when/why you choose to report data on an annual basis – see also comment about Table 1. 23. p. 14: lines 6-8. The authors claim that their study is “the first study to document the almost-perfect concordance of pharmaceutical company interests and patient organisation funding.” However, the Swedish study reported: “If a company marketed at least one drug in a disease area, there was an 83% chance that it supported a patient organisation in the disease area ($\kappa = 0.78$, 95% confidence interval: 0.66–0.90). Companies only supported patient organisations in disease areas linked to their drug portfolio.” In addition, one could argue that in order to make such a claim the authors should also look at discordances, i.e., cases where companies have an interest but do not fund the POs, like the Swedish study did. 24. p.14: line 14. References 28-30 seem to be about the profitability of orphan drugs but the statement is more general: “The almost-perfect concordance between industry interests and patient organisation activities likely reflect the commercial attractiveness of conditions targeted by pharmaceutical companies”. 25. p.14: line 14. There is a typo: “between” is repeated. 26. p.14: lines 14-17. “Such close alignment between the interests between companies and patient organisations might undermine
--	---

the credibility of patient organisations as perceived by the general public and might raise questions about patient organisations' inputs in regulatory and health technology appraisal." Please provide reference/s for these statements.

27. p. 14; lines 27-32. As noted above, if the authors want to claim that taking the "pipeline" into account makes a big difference they need to show this.

28. p. 14; lines 27-32. Again, regarding the Swedish study, because it considered a longer time-series (5 years) than this study (3 years) and, especially, because it found that "companies only supported patient organisations in disease areas linked to their drug portfolio" it is unlikely that the study, as suggested by the authors, "might have led to an underestimate of the companies' interest in some conditions." However, the authors might want to note (instead) that their study considered all companies and patient POs whereas the Swedish study only a subset. On the other hand, this study seems hampered by loads of missing data.

29. p.14; line 32-33. "robust, hierarchical matching algorithm". I would suggest reformulating this. You have not showed that it is "robust". Second, "hierarchical matching algorithm" gives the impression that you developed and tested a computer program. This comment also applies to the legend for Fig. 1.

30. p.14-15. lines 56-2. There is a big problem regarding all the analyses that concern cystic fibrosis, and which could also undermine the robustness of some of the rare disease/non-rare disease calculations. Vertex is one of those few companies that does not follow the ABPI code (see 10.1136/bmjopen-2021-053138) and thus it does not report its payments to POs. Still, it is extremely likely that Vertex is a big funder of CF patient advocacy!

31. p. 15; lines 5-9. See comment about missing payments in 2018 and 2019.

32. p. 15; line 7-8. See study by Ozieranski et al in *Sociology of Health & Illness* which looked at the share of industry contributions in POs' income.

33. p. 15. The authors do not discuss their findings about industry funding concentration. This could be put in relation to other studies. E.g. the Danish study reported (data across 6 years): "On average, the top ten donors funded 13.4 (SD = 5.2) patient organizations compared with 3.1 (SD = 2.8) patient organizations funded by the remaining 41 companies" and "On average, the top ten recipients had funding from 6.7 (SD = 3.7) drug companies compared with 2.6 (SD = 2.1) drug companies for the remaining 74 organizations."

34. p. 15; lines 11-12. It's unclear why the study findings would translate into the strong recommendation that "To minimise conflicts of interests, patient organisations should not accept payments from companies whose products they have endorsed a year before and after this endorsement."

35. p. 15; lines 15-21. These recommendations are very similar to those advanced elsewhere, e.g., the Swedish and Danish studies. See also the AbbVie example in the Danish study which is relevant to this argument.

36. p. 16. Line 23. As noted, the problem might be bigger than "underreporting". There is evidence of non-reporting or removal of reports from the public domain which need to be given very serious consideration.

37. p.16; line 24-28. Why is underreporting expected to affect all POs equally? I would expect that underreporting/non-reporting is more common among smaller drug companies and that these

	companies are different than larger companies with respect to the POs they fund. 38. There is a word missing in the last sentence in the Conclusion.
--	---

REVIEWER	Ozieranski, Piotr University of Bath
REVIEW RETURNED	06-Feb-2023

GENERAL COMMENTS	Thank you for the opportunity to review this interesting study. It is a promising piece of work but more attention is needed to several key methodological and presentational issues before it can be considered for publication. Taken altogether, some of the study's findings are difficult to interpret because insufficient detail has been provided on how the data was extracted, integrated and curated. The study would also benefit from more grounding in the existing research on POs in the UK and elsewhere. General comments Some standardisation of terminology would be important throughout the article - "funding", "transfers", "transfers of value", "TOVs" "payments" may not necessarily mean the same things. Unless you have specific reasons to use the word transfer I would recommend using the word payment as it has been used consistently by others. Keeping the terminology consistent can have important long-term benefits for this field of study. While the scope of the study is the UK the some of the key evidence (e.g. references to NICE) brought in the introduction focuses on England only. The chosen measure of the concentration of industry funding should be problematised and discussed in the context of existing research (e.g. https://pubmed.ncbi.nlm.nih.gov/36371347/) the measures typically used in this type of research, such as the Gini Index. I did not notice any description of data cleaning, integration and management procedures you might have used, which could shed important light on the structure of your data (e.g. did you standardise PO names, and how?, how did you do with duplicate entries, "split payments", any payments with negative values). These steps were a significant, and very time-consuming, part of our previous work so I'm very surprised that you did not mention them here (https://pubmed.ncbi.nlm.nih.gov/31455562/ , https://pubmed.ncbi.nlm.nih.gov/31122928/)? Very importantly, you did not reflect on the distribution of payment reports across the companies, including any cases of companies not disclosing payments to POs. This was a significant problem we and others have uncovered in previous research so I'd like to know if you came across a similar issue and how you addressed it in your study (https://bmjopen.bmj.com/content/10/9/e037351, https://onlinelibrary.wiley.com/doi/full/10.1111/1467-9566.13409). At least some of the general descriptive findings should be presented in a table presented in the main body of the article. I'm confused by how the tables are signposted in the text. For example Table 1 is not mentioned explicitly in the findings. Is it the
---

	same as Table 1 in the Supplemental Material". The same applies to Figure 2 - it is mentioned on p 10 but does not appear in the text. The tables / figures listed at the end of the manuscript lack captions which makes their interpretation difficult at times (esp. figure 3). Specific comments P 5, lines 21-28 - it would be useful if the authors specified a little bit more what they mean by the research gap they wish to target. I understand what they mean by the difference between approved drugs and drug portfolios but it would help to understand how having a "drug for certain disease" (already studied) is different from "approved drug portfolios" (apparent gap). P 5 lines 30-34 - some of the characterisation of RD POs does not differ much from how POs are characterised more broadly. A more accurate characterisation is needed to capture the uniqueness of RD POs appropriately P 7 lines 8-10 - please provide a reference to support this claim P7 lines 12-15 - there is some confusion here. Do you mean you checked / extracted data from the websites of Disclosure UK signatories or did you check / extract data from the Disclosure UK database itself (which are two completely different things)? It is important to reflect on this issue in light of our findings from this paper https://pubmed.ncbi.nlm.nih.gov/31455562/ P 7 line 32-33 (and P 5, lines 1-2) - why did you not consider a definition of patient organisations used in peer-reviewed academic work? If this is how you understand POs this definition should be provided in the Introduction. Sticking just with the EFPIA definition without any additional context can make sense methodologically but is problematic ontologically and the reader should be aware of how you define this entity from the beginning (relatedly, you only provide a very broad definition of POs in the first sentence of the article without any references to published work). P7 line 40 there is an unnecessary word in the sentence. P8 line 30-32 - what data did the verification procedure refer to? P11 - Table 1 includes the rare vs non-rate distinction but the distinction itself is only discussed later on, on p. 12, which is confusing for the reader (the section headings should be able to offer some help here) P12 line 6 - please replace transfer with funding P15 lines 12-19 - I'm not convinced by the use of the NICE study to contextualise your findings. The way you characterised it points to the issue of underreported payments and not interest alignment. P16 lines 5-9 your finding that RD POs were funded by fewer companies than non-RD POs does not seem surprising given the structure of the drug markets for most of these conditions, with very few alternative therapies available. You also mention that the share of industry funding within POs income remains unknown - this is not entirely true as this variable was examined previously by us
--	--

	and others (e.g. https://onlinelibrary.wiley.com/doi/full/10.1111/1467-9566.13409) . The fact that you did not examine it should therefore be listed as a limitation P 16 line 30 - what “similar trends” do you mean exactly here? Given the specific focus of your study on rare vs non-rare P16, line 22-23 - much more attention needs to be given to the issue of potential underreporting. There is research on this issue in the UK, Canadian and Italian context so it should be brought in here. What is the basis of your statement that you’re expecting the underreporting to affect all organisations equally? Figure 1 - please justify why you use the 2020 company report as the basis for determining companies’ interests, given that it only covers the latest data point in the payment data and is inconsistent with the approach you’re taking in relation to the clinical trial data (study start date = 01/01/2021); please define (here and in the text) what you mean by product pipeline and product portfolio. Supplemental material -  1. more detail is needed on what information exactly did you use to determine the PO condition areas. The example you used referring to PO names can be misleading as some patient organisation provide further detail on their condition areas in the “About us”, “History” or similar sections on their websites 2. More detail is needed on how you dealt with organisations dealing with more than one discrete condition area (see https://pubmed.ncbi.nlm.nih.gov/31122928/)
--	--

VERSION 1 – AUTHOR RESPONSE

#	Reviewer 1 comments	Replies
1	p. 4. The authors claim patient organisations (POs) have roles “across all stages of drug development, approval and access”. However, it would be easy to find examples of stages where POs do not have roles, e.g., related to company decisions, marketing, and pricing. I suggest to reformulate this.	Many thanks for your comment. We have amended the text to reflect your comment (p.4, lines 16-17). However, while we agree that patients do not take part in companies' decisions on which disease areas to invest in - as these usually involve in-house estimates of ROI, existing know-how and expertise - we are not sure that marketing and pricing fall under the "stages of drugs development".
2	p. 4. Some additional studies of industry funding of POs could be cited here to underscore the international nature of this literature, including from Canada, Australia, Finland and Denmark.	Many thanks for your comment. We have amended the text to reflect the wider geographical scope of the research conducted so far (p. 4, line 22).

3	p.4; lines 31-40. I believe the authors downplay some of the literature here. They claim that “whether companies fund patient organisations operating in therapeutic areas relevant to companies’ approved drug portfolios and research and development pipelines remains unanswered.” However, the Swedish study cited earlier in this paragraph looked at whether companies fund patient organisations operating in therapeutic areas relevant to the companies’ drug portfolios (see Table 6 in that paper). It is true that the study did not look at R&D pipelines, but it’s unclear if this makes any difference (see more below), and, moreover, the Swedish study looked at funding over a longer time-period (2014-2018), meaning some of the “research and development pipelines” had become part of “approved drug portfolios”.	Thank you for your valuable feedback. We fully concur with the reviewer's suggestion to provide more context for the existing literature in our paper, and we have made several changes to various sections to address this concern (p. 4, lines 23-32). Most importantly, we have highlighted the geographical scope of our analysis (i.e., the UK) and emphasized the added value of our paper, which is our systematic investigation of companies’ R&D pipelines, making it a novel contribution to the literature. As the reviewer pointed out, the development and testing of a drug can take a significant amount of time, and our inclusion of the companies' pipelines, in addition to their portfolios, allows us to better capture companies’ commercial interests.
4	p. 4. The distinction rare/non-rare disease is central to this paper. However, the authors could do more to justify this through a more detailed review of the literature, and, based on this, develop some preliminary hypotheses or expectations. For example, there is some research suggesting that it might be more difficult for rare-disease POs to get non-industry funding (mentioned in the Discussion). Also, in Denmark, POs with fewer members (less than 100) relied more on industry funding, see 10.1016/j.healthpol.2022.11.003	Many thanks for your comment. We have considerably expanded our argument on what makes rare disease advocacy different than non-rare one in response to another comment (comment 2 from Reviewer 2, referring to p. 5, lines 30-34). In the revised paragraph (p. 4-5 , lines 35-15) we added details about the specific nature of rare diseases, their important advocacy role and involvement in appraisal processes. We also discuss the specific role played by rare disease focused patient organisations in generating medical knowledge and informing regulatory decisions. Furthermore, we have added preliminary hypotheses for our study based on existing evidence (p. 5-6, lines 31-4). After reviewing the survey mentioned in the Danish paper, we have decided not to use this as part of our hypothesis formulation. Firstly, the survey highlights how POs with less than 1,000 (not 100)

		rely more on industry funding. This is still a very high number for rare disease patient organisations, that sometimes are only few hundreds in the UK, so we did not feel comfortable making this assumption. Finally, we have further clarified our objectives (p. 5, lines 24-32) as:  1. Analysing the volume and value of payments to patient organisations; 2. Characterising the financial relationships between industry and patient organisations focusing on rare vs non-rare diseases in the UK; 3. Evaluating the concordance between companies' commercial interest and patient organisations' activities; 4. Examining the concentration of industry funding, namely how many companies funded each patient organisation, and the extent to which organisations might have been reliant on funding from a single company.
5	p. 5; line 4-6. The authors might also want to check out the Danish study for this statement as it similarly addressed the question of concentration of funding.	Thank you for your comment. We have now incorporated the paper on the Danish context of this issue in our analysis (p. 5, line 16-23).
6	p. 6; line 17-18. It is the PMCPA that sanctions companies, not the ABPI. The ABPI Board may suspend or expel a company but such cases are extremely rare.	Thank you, that is indeed correct. Despite the Code of Practice is for ABPI members, it is enforced by the PMCPA. We have amended the text accordingly (p. 7 , lines 11-12).
7	p. 6: line 20-22. Why would companies report payments to UK POs in other currencies?	Many thanks for your comments. Despite being a minority, some payments were reported in EUR, USD, CHF, SEK, NKK and DKK. This was the case when payments were - for example - from headquarters, or companies based outside of the UK invoice payments in their currency

8	p. 7: line 17 & Fig. 1. a. Why did you restrict the study start date in Clinicaltrials.gov to 1/01/2021 when the funding you study took place between 2018-2020? Would it not have been more relevant to include trials running between 2018 and 2020, or even earlier, given the time it takes to develop and test a new drug? b. Related to the previous point: the authors should provide data on the results from their searches, e.g., add actual numbers in Figure 1. c. I think it would be helpful for others (and increase the relevance of the study) if the authors could also add a supplementary file with company interests across disease areas. d. It was unclear to me if the analysis also evaluated cases where companies did not fund POs in diseases linked to their commercial interests, as the Swedish study did, see comment above. If not, why?	Thank you for your helpful comment. We have addressed this comment fully in the following ways: a. We have included studies with a start date no later than 1/01/2021. This encompasses the period suggested by the reviewer (2018-2020). As suggested by the reviewer the comment, drug R&D is a lengthy process and pharmaceutical companies are likely to be interested in the patients affected by a certain disease prior to commercialising a drug. It is a fair assumption that trials starting before 2018 and after 2020 would still be of potential interest to our study. We have amended the text in Figure 1 to make it clearer. b. Many thanks for your comment. We have now added a figure in the Supplemental Materials (Figure 1) to show the source of the identified interest (e.g. annual report, company's website or ClinicalTrials.gov). c. Many thanks for your comment. We have added a table indicating companies' interest by ICD-11 codes in the Supplemental Materials (Table 5). However, companies' interests were screened opportunistically only in disease areas where they made a payment to a specific patient organization, and therefore this table should not be considered exhaustive. d. Many thanks for your comment. In our analysis, our starting point were the payments of industry to patient organisations rather than their pipelines/portfolio (i.e. we did not focus on payments that were not made, regardless of the company's interest). Furthermore, we determined the degree of concordance rather than whether there is one (binary approach), which made exploring that aspect even more complicated. In fact, we considered companies as definitely
---	--	--

		having an interest in a specific condition only if the remit of the patient organisation was as specific as the clinical indication of the company's product(s). Other more general instances were coded as maybe yes. Therefore, a table like the one presented in the Swedish study (Table 6.), where the ten top industry donors were matched with the top funded disease areas could not be produced as the ICD-11 levels were not homogeneous.
9	p.9; line 7. The use of the word “significantly” might suggest to some that the change was statistically significant.	Thank you very much for your comment. We have changed significantly to substantially in the text (p. 10, line 3).
10	p.9; line 4. Most important comment: The substantial increase in funding in 2020 seems remarkable at first sight. I would have expected reduced funding of POs during Covid. However, what I think is going on is that the increase is an artifact of company reporting practices (or, alternatively, of the authors' search strategy). Indeed, the number of reporting companies in 2018 is about 30 and in 2020 it is about 60 (see Supp Fig.1)! This strongly suggest that the study has major missing data problem which challenges the validity of the results. It is very surprising that the authors choose not to foreground and comment on this (here as well as in the discussion) and interpret all their calculations and findings in light of this very likely bias. E.g., it does not seem credible that AbbVie made no payments in 2018! Also, because non-reporting seems more common among smaller companies, which might be more likely to focus on rare diseases, it probably undermines the comparison between rare/non-rare, in addition to calculations of funding concentration. There are a number of studies that have looked at company reporting practices, including in the UK, as well as a recent Nordic comparative study which suggested that countries without centralized databases had higher levels of under- and non-reporting. The authors should re-evaluate their study in light of this missing data problem.	Thank you for your comment. While we find the assumption regarding the funding trends during the initial stages of Covid-19 interesting, we could not support such statement with any evidence from the peer-reviewed literature. Based on our observations of the data, we find that quite a lot of payments were earmarked as Covid-19 related support (in our data set 25.4% of all payments in 2020 included the word "pandemic", "Covid-19", "coronavirus" etc). Regarding the potential missing data problem, we acknowledge that the increase in the number of reporting companies from 37 in 2018 to 60 in 2020 could be due to underreporting in previous years. However, we believe that this is only part of the explanation. When we considered the subset of companies that disclosed consistently over the three years in our sample (n=37), we found that the value of payments they disclosed comprised 93% and 86% of the total value of payments for 2019 and 2020, respectively. Furthermore, a similar upward trend in the number and value of payments was observed in the UK

		between 2012 and 2016, as shown in a BMJ paper from 2019 co-authored by the two reviewers. We have added a comment on this issue (p. 10, lines 4-7) and a new table in the Supplemental Material (Table 4) to clarify this point. While we agree that it is unlikely that big companies such as AbbVie did not make payments in years for which there are no disclosures available, it is commonly assumed in the literature that a missing report could indicate either no payment or failure to disclose (for example, see the BMJ paper from 2019 co-authored by the two reviewers “A missing report may indicate no payments or a failure to disclose”). We have also included this as an assumption in our analysis (p. 7, lines 15-17) and have further highlighted it in the discussion of our limitations (p. 16-17, lines 31-7). Lastly, we would like to address the assumption of a disproportionate effect of underreporting on rare disease focused POs. We have doublechecked our sample and smaller pharmaceutical companies which are known for focusing on rare disease-related products are in fact present in our analysis, with Vertex being the only major one missing. Also, as can be seen from Figure 3, many large companies have an interest in rare diseases (e.g. Sanofi, Pfizer, Novartis). Furthermore, we found no evidence in the peer-reviewed literature regarding the higher likelihood of smaller companies marketing rare disease focused products to underreport compared to larger companies. In light of this, the negative effect of a centralized database which the two reviewers have studied extensively should not depend on the prevalence of the disease.
--	--	--

11	p.9; lines 5-7. Could you provide more information on specific diseases? That is, provide a breakdown of the broad disease areas (level 1) into their lower-level diseases (level 2 or below). This might allow you to make more granular interpretations of funding patterns and compare with other studied countries. Some information is provided later on p. 11 which could be moved here and expanded.	Thank you for your comment. Table 1 in the supplementary materials provides a breakdown of how we considered ICD-11 levels (retrieved from https://icd.who.int/browse11/l-m/en) to code the interest variable. As mentioned in an earlier comment, we do not believe different levels would be comparable as (1) not all disease areas have the same number of levels (1 to 5) - please see linearisation map provided as part of the supplemental materials; (2) e.g. funding to acute lymphocytic leukemia would not be comparable to funding to cancer in general, and (3) while interesting, we believe it is out of the scope of the present study.
12	p. 9; line 11. I could not identify Supp Table 1 for this. Do you mean Supp Table 2?	Thank you for spotting this error! We actually referred to Table 1 in the main document. We have changed the text accordingly.
13	p.9; lines 14-15. What would the results have been if you would only consider the “portfolio”? In other words, what difference does the “pipeline” make? And how much difference did the inclusion of trials make? This is important because you later claim that an advantage of this study is that it included the “pipeline”. See also comment above.	Thank you for your comment. We initially tried to quantify this explicitly. However, it turned out to be quite complex. A static breakdown between pipeline and portfolio is difficult in a retrospective study since we were observing companies’ websites in 2022 rather than when the payment was made. However, as the reviewer noted in a previous comment, drug development can take a substantial amount of time. We believe that conditions that are represented in the pipeline but perhaps not yet in the portfolio provide us with a better overview of the commercial interests of companies.
14	Table 1: consider not breaking down results per year. There are no obvious differences between years (other than the total sums).	Thanks for your suggestion. We have changed the table in the main document (now Table 2) with the overall results (i.e. not broken down by year). However, we have kept the original table in the Supplemental Materials (Table 8)

		as, while this is fine for percentages, we think it is important to compare absolute values among the same sample (i.e. as noted previously, not all companies disclose payments for all the years in analysis).
15	Fig 3. "Figure 3 shows therapeutic areas in order from most to least funded". This does not seem to be the order in the figure.	Thank you for your comment. We have updated Figure 3 to be in decrescent order.
16	Fig 3. is slightly confusing because the order or rare/non-rare is different for different therapeutic areas (also Fig S2).	Thanks. We have changed Figure 3 to a stacked bar for ease of interpretation.
17	p. 11: lines 6-7: "From 2018 to 2020, the transfer to patient organisations targeting rare diseases increased more compared to those focusing on more prevalent conditions (80% vs 57%)." Could this be explained in part by less underreporting by smaller companies targeting rare diseases?	Thanks for your comment. As discussed in earlier responses, underreporting is expected to affect all analyses leveraging disclosure data from pharmaceutical companies, and this is no exception. However, as we have extensively reflected in the discussion (p. 16-17, lines 31-7) and in earlier responses (see response to comment 10), we do not believe this poses a significant risk to the validity of our analysis.
18	p.11; lines 16-19. See comment 11.	Please refer response to comment 11.
19	p.11; lines 16-19. The reference to Table 2 does not seem right here. Table 2 shows specific POs not disease areas.	Thanks for noticing this. We erroneously added a reference to Table 2, which has now been removed.
20	Please re-check Table 2. I was surprised that single transfers would be this dominating. For example, according to Table 3, Chiesi made one and only one transfer to the CF Trust worth 1.3 million. However, I quickly checked Chiesi's reports and the company had 3 payments in 2019. Another example: Epilepsy Society reportedly had two industry funders but Table 3 also reports that one payment accounted for 100% of the funding, which must be inaccurate.	Many thanks for picking up on the inconsistency. In the original table, Highest transfer referred to the cumulative transfers by the top funder, while the Share highest transfer/ overall funding was actually referring to the Top funder share of overall payments (i.e. a measure of company dependence). We have now amended the table, but we decided to move it to the Supplemental Materials (Table 7) to make space for a table of more generals payments statistics, as per comment G6 from Reviewer 2.

21	p. 13: lines 6-10. Consider having the IQR values as %.	Thank you for your comment. As suggested by the reviewer, we have changed IQR values as percentages in the text (p. 14, lines 1-19).
22	p. 13: lines 11-17. Why did you choose to calculate/report this per year rather than (also) across all 3 years? In general, it is unclear when and why you choose to report data across all years and when/why you choose to report data on an annual basis – see also comment about Table 1.	Many thanks for your comment. In this paragraph we refer to results for all years, and just add a sentence to provide more granularity in terms of yearly trends. In response to your earlier comment on Table 1, we have kept references to overall values in the main document and moved results by year to the Supplemental Materials.
23	p. 14: lines 6-8. The authors claim that their study is “the first study to document the almost-perfect concordance of pharmaceutical company interests and patient organisation funding.” However, the Swedish study reported: “If a company marketed at least one drug in a disease area, there was an 83% chance that it supported a patient organisation in the disease area ($\kappa = 0.78$, 95% confidence interval: 0.66–0.90). Companies only supported patient organisations in disease areas linked to their drug portfolio.” In addition, one could argue that in order to make such a claim the authors should also look at discordances, i.e., cases where companies have an interest but do not fund the POs, like the Swedish study did.	Thank you for your comment. We have changed both the introduction - where we indicate the existing gaps in the literature (p. 4, lines 23-32) - as well as the text in the discussion (p. 15, lines 3-5) to reflect this point.
24	p.14: line 14. References 28-30 seem to be about the profitability of orphan drugs but the statement is more general: “The almost-perfect concordance between industry interests and patient organisation activities likely reflect the commercial attractiveness of conditions targeted by pharmaceutical companies”.	Thanks for picking on the reference mismatch, that dated a previous version of the paper! We have updated the references to reflect the non-rare-specific statement.
25	p.14: line 14. There is a typo: “between” is repeated.	Thanks, we have removed the redundant word from the text.
26	p.14: lines 14-17. “Such close alignment between the interests between companies and patient organisations might undermine the credibility of patient organisations as perceived by the general public and might raise questions about patient organisations’ inputs in regulatory and health technology appraisal.” Please provide reference/s for these statements.	Many thanks for your comment. We have added two references to substantiate the statement made (Rose et al., 2017 and McCoy et al. 2017).

27	p. 14; lines 27-32. As noted above, if the authors want to claim that taking the “pipeline” into account makes a big difference they need to show this.	Thank you for your comment. Please refer to our earlier responses (e.g. to comment 3) where we explain why we believe our systematic investigation of companies’ R&D pipelines represents a novel contribution to the literature.
28	p. 14; lines 27-32. Again, regarding the Swedish study, because it considered a longer time-series (5 years) that this study (3 years) and, especially, because it found that “companies only supported patient organisations in disease areas linked to their drug portfolio” it is unlikely that the study, as suggested by the authors, “might have led to an underestimate of the companies’ interest in some conditions.” However, the authors might want to note (instead) that their study considered all companies and patient POs whereas the Swedish study only a subset. On the other hand, this study seems hampered by loads of missing data.	Thank you for your thoughtful comment and suggestion about the contextualisation of our study in the wider literature. We have revised the manuscript in light of your suggestions (p. 28, lines 28-34).
29	p.14; line 32-33. “robust, hierarchical matching algorithm”. I would suggest reformulating this. You have not showed that it is “robust”. Second, “hierarchical matching algorithm” gives the impression that you developed and tested a computer program. This comment also applies to the legend for Fig. 1.	Thank you for your valuable feedback. We agree that robust is subjective and difficult to quantify. Therefore, we have revised the manuscript to use the term replicable instead, to convey that our methodology can be reliably reproduced. In addition, we have changed hierarchical matching algorithm to classification model, to avoid any confusion (p. 15, lines 32-34). However, we would like to clarify that the term algorithm is not limited to computer science, and can be used more broadly to refer to a set of rules or processes that map the way to a solution. Also, we hope that the terminology now used (i.e. model) does not raise the same issues, as it could also be associated to mathematical analysis.
30	p.14-15. lines 56-2. There is a big problem regarding all the analyses that concern cystic fibrosis, and which could also undermine the robustness of some of the rare disease/non-rare disease calculations. Vertex is one of those few companies that does not follow the ABPI code (see 10.1136/bmjopen-2021-053138) and thus it does not report its payments to POs. Still, it is extremely likely that Vertex is a big funder of CF patient advocacy!	Thank you for bringing up this point. We acknowledge that the lack of mandatory reporting of payments by pharmaceutical companies to patient organizations is a major limitation in this field, which is likely to bias any analysis. Specifically, we agree that Vertex is one of the companies that does not follow the ABPI code and therefore does not

		report its payments to POs, but is likely a big funder of cystic fibrosis patient advocacy. We have updated the manuscript to include this limitation in our discussion (p. 17, lines 31-41), and to highlight that the absence of Vertex in our sample may lead to an underestimation of the total funding for CF. As we emphasize in our discussion, we are not focusing on absolute values (£), but rather on the relative funding levels of rare diseases in general, and CF in particular. Our analysis demonstrates that CF is one of the most well-funded rare diseases, and our updated discussion points out that this is likely an underestimate given the lack of Vertex in our sample.
31	p. 15; lines 5-9. See comment about missing payments in 2018 and 2019.	Many thanks for your comment. Please refer to our response to comment 10 and our revised discussion and limitations sections (p. 16-17, lines 31-7).
32	p. 15; line 7-8. See study by Ozieranski et al in Sociology of Health & Illness which looked at the share of industry contributions in POs' income.	Many thanks. We initially referred to our sample, but have now added evidence from the literature and rephrased the text in the main document accordingly (p. 17, lines 5-9).
33	p. 15. The authors do not discuss their findings about industry funding concentration. This could be put in relation to other studies. E.g. the Danish study reported (data across 6 years): "On average, the top ten donors funded 13.4 (SD = 5.2) patient organizations compared with 3.1 (SD = 2.8) patient organizations funded by the remaining 41 companies" and "On average, the top ten recipients had funding from 6.7 (SD = 3.7) drug companies compared with 2.6 (SD = 2.1) drug companies for the remaining 74 organizations."	Many thanks for your comment. We have now added a paragraph in the discussion reflecting on our results and comparing it with the existing literature (p. 16, lines 9-15).
34	p. 15; lines 11-12. It's unclear why the study findings would translate into the strong recommendation that "To minimise conflicts of interests, patient organisations should not accept payments from companies whose products they have endorsed a year before and after this endorsement."	Thank you for your comment. We have amended the text and softened our recommendation which still relates to mitigating conflicts of interest (p. 16, lines 16-19). We understand that not accepting payments from

		pharmaceutical companies that might have commercial interests in the relevant condition supported by the PO would be financially unattainable and might minimize the important contributions POs make to the R&D process. Our aim with this recommendation was to reflect the importance of maintaining the integrity of POs.
35	p. 15; lines 15-21. These recommendations are very similar to those advanced elsewhere, e.g., the Swedish and Danish studies. See also the AbbVie example in the Danish study which is relevant to this argument.	Thank you for your feedback. Our paper presents novel findings while building on the existing literature, and it is not unexpected that our recommendations regarding the importance of improving transparency and addressing policy implications are similar to those of previous studies. We have taken your comment regarding the dominance of certain funders and their potential negative impact on public health concerns into consideration and added a sentence in the main document to address this point (p. 16, line 23-26).
36	p. 16. Line 23. As noted, the problem might be bigger than “underreporting”. There is evidence of non-reporting or removal of reports from the public domain which need to be given very serious consideration.	Many thanks for your comment. We have amended the text to reflect the impact that underreporting and missing data could have on our analysis (p. 16-17, lines 31-7).
37	p.16; line 24-28. Why is underreporting expected to affect all POs equally? I would expect that underreporting/non-reporting is more common among smaller drug companies and that these companies are different than larger companies with respect to the POs they fund.	Many thanks for your comment. Our initial sentence intended to reflect the fact that underreporting is likely to be assigned at random and not linked to the rarity of diseases, as we have not come across any publication hinting otherwise. Also, while it is true that there are a few rare-focused companies, much funding also comes from larger companies (e.g. Sanofi). However, we have expanded the discussion on underreporting and non-reporting issues and taken into account your feedback by deleting the sentence you highlighted (p. 16-17, lines 31-7).

38	There is a word missing in the last sentence in the Conclusion	Thanks - we have fixed that!
#	Reviewer 2 comments	Replies
General comments		
G1	Some standardisation of terminology would be important throughout the article - "funding", "transfers", "transfers of value", "TOVs""payments" may not necessarily mean the same things. Unless you have specific reasons to use the word transfer I would recommend using the word payment as it has been used consistently by others. Keeping the terminology consistent can have important long-term benefits for this field of study.	Thank you for your comment. We have changed the terminology throughout the text accordingly. We now use "payment" and "funding" to refer to the financial relationship between patient organisations and industry as suggested by the reviewer and the peer-reviewed literature.
G2	While the scope of the study is the UK the some of the key evidence (e.g. references to NICE) brought in the introduction focuses on England only.	Thanks for your comment. We have now added a reference to the PACE process in Scotland to reflect the relevance of the analysis for the entire UK and not England only (p. 5, lines 7-10).
G3	The chosen measure of the concentration of industry funding should be problematised and discussed in the context of existing research (e.g. https://pubmed.ncbi.nlm.nih.gov/36371347/) the measures typically used in this type of research, such as the Gini Index.	Thank you for your comment. The use of a Gini coefficient would have been an interesting approach as adopted in your paper looking at inequalities of funding to HCOs. However, as reflected in the text now (p. 9, lines 2-8) and as per the discussion in your analysis, presenting multi-year comparisons using the Gini coefficient presents several methodological challenges. We have, therefore, used descriptive statistics to report on the funding concentration instead.
G4	I did not notice any description of data cleaning, integration and management procedures you might have used, which could shed important light on the structure of your data (e.g. did you standardise PO names, and how?, how did you do with duplicate entries, "split payments", any payments with negative values). These steps were a significant, and very time-consuming, part of our previous work so I'm very surprised that you did not mention them here (https://pubmed.ncbi.nlm.nih.gov/31455562/ , https://pubmed.ncbi.nlm.nih.gov/31122928/)?	Many thanks for your comment. We have indeed undergone an extensive cleaning and coding process that we have now explained in greater detail. You can now find all the details in the supplemental materials. We have added (1) the Excel file containing all payment information (company making the payment, name of receiving PO, details of payment, year, currency and value of payment); (2) the Excel file containing aggregated payment data, with details on patient

		organisations, disease areas, company interest and additional inputs used in the analyses (e.g. inflation and exchange rates) and (3) further details on variables coding and disclosure payments details. Finally, please note that all these data will be made available via the BMJ Open.
G5	Very importantly, you did not reflect on the distribution of payment reports across the companies, including any cases of companies not disclosing payments to POs. This was a significant problem we and others have uncovered in previous research so I'd like to know if you came across a similar issue and how you addressed it in your study (https://bmjopen.bmj.com/content/10/9/e037351 , https://onlinelibrary.wiley.com/doi/full/10.1111/1467-9566.13409).	Thank you for your comment. We have now discussed this as part of the limitations (p. 16-17, lines 31-7) and added a two tables in the Supplemental Materials (Tables 3 & 4) regarding the consistency of companies' reports over time.
G6	At least some of the general descriptive findings should be presented in a table presented in the main body of the article.	Many thanks for your comment. We have now added a new table (Table 1) presenting some general descriptive findings and moved Figure 2 in the previous version of the manuscript to the supplemental materials.
G7	I'm confused by how the tables are signposted in the text. For example Table 1 is not mentioned explicitly in the findings. Is it the same as Table 1 in the Supplemental Material". The same applies to Figure 2 - it is mentioned on p 10 but does not appear in the text.	Thanks for noticing. We have amended the in-text caption of Table 1 as per an earlier comment.
G8	The tables / figures listed at the end of the manuscript lack captions which makes their interpretation difficult at times (esp. figure 3).	We are unsure about this comment, as captions are mentioned at the end of the paper. They can be found after the references as per journal requirements.
Non-general comments		
1	P 5, lines 21-28 - it would be useful if the authors specified a little bit more what they mean by the research gap they wish to target. I understand what they mean by the difference between approved drugs and drug portfolios but it would help to understand how having a "drug for certain disease" (already studied) is different from "approved drug portfolios" (apparent gap).	Thank you for the comment – we have amended the text to better reflect the novelty of our paper relative to the existing literature (p. 4-5, lines 23-23). We have specified the difference between our analysis and the existing literature and in particular a study conducted by the two reviewers looking at a similar issue in Sweden. This includes a different geographical focus

		(namely the UK) and a novel analysis of R&D pipeline rather than just marketed drug portfolios.
2	P 5 lines 30-34 - some of the characterisation of RD POs does not differ much from how POs are characterised more broadly. A more accurate characterisation is needed to capture the uniqueness of RD POs appropriately	Many thanks for your comment. We have considerably expanded our argument on what makes rare disease advocacy different than non-rare one, which we hope addresses the point you raise. In the revised paragraph (p. 4-5 , lines 33-15) we added details about the specific nature of rare diseases, their important advocacy role and involvement in appraisal processes. We also discuss the specific role played by rare disease focused patient organisations in generating medical knowledge and informing regulatory decisions.
3	P 7 lines 8-10 - please provide a reference to support this claim	Many thanks for your comment. This sentence reflects email correspondence with Heidi Graham, Disclosure Manager from ABPI. More specifically, this was her reply to our inquiry of why there were non-ABPI members that disclosed payments to patient organisations, as that seemed to be a requirement for ABPI members only and it is not required by UK law. However, the Industry-complex paper, reads "Importantly, while no precise estimate of the total number of companies is available, 'virtually all pharmaceutical companies operating in the UK' (ABPI, 2019, p. 6) follow the ABPI Code.". Therefore, we referenced the 2021 ABPI Code of Practice.
4	P7 lines 12-15 - there is some confusion here. Do you mean you checked / extracted data from the websites of Disclosure UK signatories or did you check / extract data from the Disclosure UK database itself (which are two completely different things)? It is important to reflect on this issue in light of our findings from this paper https://pubmed.ncbi.nlm.nih.gov/31455562/	Thanks for your comment. We checked websites of all companies abiding by the ABPI code and retrieved data from Disclosure UK as well as single websites. We have now amended the text (p. 7, lines 12-15) and added Supplemental Materials (p. 1, lines 12-14) to provide more clarity and details on the websites screened.

5	P 7 line 32-33 (and P 5, lines 1-2) - why did you not consider a definition of patient organisations used in peer-reviewed academic work? If this is how you understand POs this definition should be provided in the Introduction. Sticking just with the EFPIA definition without any additional context can make sense methodologically but is problematic ontologically and the reader should be aware of how you define this entity from the beginning (relatedly, you only provide a very broad definition of POs in the first sentence of the article without any references to published work).	Thank you for your helpful comment. After a careful and thorough review of the peer-reviewed literature, we established that the EFPIA definition matches the one given in the academic literature. We have included relevant references (p. 7-8, lines 32-1).
6	P7 line 40 there is an unnecessary word in the sentence.	Thanks, we have corrected that in the text.
7	P8 line 30-32 - what data did the verification procedure refer to?	Thank you for your helpful comment. We have now amended the text to reflect that the validation process was followed for all data sources described throughout the section (p. 8, lines 37-40).
8	P11 - Table 1 includes the rare vs non-rare distinction but the distinction itself is only discussed later on, on p. 12, which is confusing for the reader (the section headings should be able to offer some help here)	Thank you for your comment. We have now added an additional row referring to overall results, irrespective of the disease rarity (Table 2). Furthermore, the comparative analysis between rare and non-rare POs was introduced in the methods section, so we do not believe the reader will find it confusing as long as provided with the general results that we have added to the table.
9	P12 line 6 - please replace transfer with funding	Thank you for your comment we have replaced transfers with payments throughout the text as per your general comment!
10	P15 lines 12-19 - I'm not convinced by the use of the NICE study to contextualise your findings. The way you characterised it points to the issue of underreported payments and not interest alignment.	Thank you for your comment. As suggested we have brought forward a discussion point made by the authors of the NICE study (p. 15, lines 19-20). Namely, we emphasize that due to the uncertainty about the financial ties between patient organisations and industry, the formers contribution to NICE hearings can be seen as compromised with concerns of conflict of interest.
11	P16 lines 5-9 your finding that RD POs were funded by fewer companies than non-RD POs does not seem surprising given the structure of the drug markets for	Thanks for your comment. We are familiar with your paper on financial

	most of these conditions, with very few alternative therapies available. You also mention that the share of industry funding within POs income remains unknown - this is not entirely true as this variable was examined previously by us and others (e.g. https://onlinelibrary.wiley.com/doi/full/10.1111/1467-9566.13409) . The fact that you did not examine it should therefore be listed as a limitation	dependency. We initially referred to our sample, but have now added evidence from the literature and rephrased the text in the main document accordingly (p. 15, lines 5-9).
12	P 16 line 30 - what "similar trends" do you mean exactly here? Given the specific focus of your study on rare vs non-rare	Thank you for your comment. The similar trends we referred to are the most funded diseases and the rough estimates of payments reported in pounds, which are aligned reported by previous studies. These results increase confidence on the robustness of the data used in the analysis. We have revised the sentence to make this clearer (p. 17, lines 11-14).
13	P16, line 22-23 - much more attention needs to be given to the issue of potential underreporting. There is research on this issue in the UK, Canadian and Italian context so it should be brought in here. What is the basis of your statement that you're expecting the underreporting to affect all organisations equally?	Thank you for your feedback. While our initial sentence aimed to suggest that underreporting is not expected to be linked to the rarity of diseases, we recognize the importance of acknowledging that there may be other factors affecting underreporting. To address this, we have expanded our discussion on under and non-reporting issues in the revised version of the manuscript (p. 16-17, lines 31-7). As a result, we have decided to delete the aforementioned sentence.
14	Figure 1 - please justify why you use the 2020 company report as the basis for determining companies' interests, given that it only covers the latest data point in the payment data and is inconsistent with the approach you're taking in relation to the clinical trial data (study start date = 01/01/2021); please define (here and in the text) what you mean by product pipeline and product portfolio.	Thanks for your comment. We have included studies with a start date no later than 1/01/2021. We have amended the text in Figure 1 to avoid confusions. We define pipeline as the collection of drug candidates being developed by a pharmaceutical company, at various stages of development, from preclinical research to clinical trials. Conversely, portfolio refers to a group of drugs that a pharmaceutical company has already developed, gained regulatory approval for, and is actively marketing or selling. We

		have added these definition in the text too (p. 8, lines 12-16).
15	Supplemental material - 1. more detail is needed on what information exactly did you use to determine the PO condition areas. The example you used referring to PO names can be misleading as some patient organisation provide further detail on their condition areas in the "About us", "History" or similar sections on their websites 2. More detail is needed on how you dealt with organisations dealing with more than one discrete condition area (see https://pubmed.ncbi.nlm.nih.gov/31122928/)	Many thanks for your comment. We have made available the ICD-11 linearisation map we used for coding therapeutic areas as well as the Excel file with complete data on therapeutic areas. In a nutshell, patient organisations were coded both generally (ICD-11 level 1, most general) as to the most specific available (see linearisation map). Regarding to your comment on the patient organisations name, we are unsure about what you are referring to, as in our text we do not discuss basing our categorisation on the organisations' names but rather using their description in their website (i.e. Blood Cancer UK says in their website in the "About us" section, that their mission is to beat blood cancer, therefore, we used that as disease area). Regarding the situation when companies focused on multiple we listed all diseases targeted as mentioned on the website (e.g. Asthma + Lung UK was coded as targeting ICD-11 12 Diseases of the respiratory system AND the following conditions: asthma, lung disease). We hope that this explanation along the full dataset will shed some light on our approach.

VERSION 2 – REVIEW

REVIEWER	Mulinari, Shai Lunds Universitet, Sociology My partner is employed by ICON, a global Contract Research Organization whose customers include many pharmaceutical companies
REVIEW RETURNED	23-Mar-2023
GENERAL COMMENTS	Thank you for the opportunity to re-review this manuscript. I believe the authors have made important improvements to their study, especially by more precisely explaining how it contributes the existing literature. I also appreciate the additional information they present as well as the efforts to respond to my comments – most of which are now addressed.

However, I still think the authors do not properly and sufficiently address the issue of missing data. The authors say in the Methods: "If payments were not disclosed in Disclosure UK nor in the company's website, we assumed the company was (sic) did not make any payments to patient organisations in a given year which is commonly assumed in the literature." However, in their rebuttal letter they quote from the UK study referenced in this sentence as following: "A missing report may indicate no payments or a failure to disclose". Their statement in the paper is thus incomplete. Indeed, the literature is very clear on the point that lack of payment reports from companies is more likely to represent a failure to disclose, or a removal of reports from the public domain.

What strongly indicates that they are dealing with the latter (i.e., missing data) is not only the increase in the number of companies reporting but also (1) that there are apparently no companies that reported in 2020 but not in 2018 or 2019, and (2) the number of companies reporting in 2018 and 2019 are fewer than what was reported in the previous UK study (as well as in the much smaller Nordic countries).

In addition, the authors chose not to comment on the fact that the previous UK study, which included data from 2012-2016, found reported payments in 2015 and 2016 that were larger in total than those reported for 2018 in this study; and the value in 2016 was almost £21 million, compared to the £18 million in 2020 reported by the authors. Instead, they refer to the average across all years, 2012-2016, in the previous UK study which "erases" the higher values that might complicate their argument. Overall, the claim (hypothesis 1) that the present "results confirm our expectations of increasing industry funding as expressed in Hypothesis 1" seems very shaky, therefore, as payments reported in 2016 were larger than what is reported here for 2018-2020.

The authors main counter-argument is most likely that there was an increase in payments among the companies that disclose consistently between 2018-2020. However, without having a longer time series it is difficult to know if this reflects "random" fluctuations or is a "real" effect.

I suggest the authors delete this analysis and hypothesis 1 from their study.

Yet, the fact that they seem to have "complete" data from 37 companies immediately raises the question what happens if the authors only include those companies in their analysis? The authors make the following claim in their limitation section:

"While this might bias our results, the impact of this was considered to be limited. Most notably, despite the differences in sample size, absolute values of payments are very similar when considering only companies that consistently disclosed payments across years (n=37). For example, in 2020, payments from those companies that disclosed consistently across the study period amounted to £15.5 million versus £18 million when any payment disclosed in that year is considered (86%)."

	The statement that the “impact of this was considered to be limited” is not really justified but can be tested by running the analyses with the 37 companies instead. This should not be very problematic since the value of payments from these firms comprise 93% and 86% of the total value of payments for 2019 and 2020, respectively. I suggest the authors run their analyses (especially statistical tests) with the 37 companies instead. Alternatively, they could run the analysis with 2020 data only.
--	--

REVIEWER	Ozieranski, Piotr University of Bath
REVIEW RETURNED	30-Mar-2023

GENERAL COMMENTS	I do appreciate the opportunity to review the revised paper, which has improved significantly. Thank you for responding to my comments. The paper can be a very important and timely intervention which expands on the literature in several key ways, but before it can be considered for publication a number of outstanding important issues need looking into. I have noted my comments below. Strengths and limitations of this study The fact that companies disclosing consistently had similar payment patterns does not allow you to make inferences about non-disclosing companies. The pattern and composition of payments can be determined by many variables which you are not controlling for. To use an example from a related field, company size and focus on narrow drug portfolios seem to be strongly associated with a high share of R&D payments. Main body of the article P5 line 2 The first sentence could be made more precise (e.g. some patient organisations also represent and support the needs of carers) and supported by references. Main points from this sentence are also repeated below in para 1 and 2, which makes at least some of it redundant. P5 line 3 In the sentence with the definition of patient organisations please be specific about its source. If it is an industry definition, this must be made clear to the reader. P5 line 33 - a definition of rare diseases would be helpful as they vary across jurisdictions P5 line 35 - what do you mean by “fragmented”? P5 line 22-3 - for further context, a recently published study has considered concentration of payments in UK countries looking jointly at HCOs and POs https://bmjopen.bmj.com/content/13/3/e061591.full P6 H3 - this hypothesis is not entirely convincing given the already well-documented process of “orphanisation” of drug development and the fact that many companies use the drug discovery model established in the field of rare diseases in other fields. Given this, companies may, in fact, have more incentives to support RD POs
--

	over non-RD POs. You may agree or disagree with this suggestion but it is important to foreground the hypothesis appropriately in the existing research (i.e. the processes of orphanisation, salami-slicing of diseases etc). P8 para 1 - it would be important to inform the readers about the 3-year mandatory data retention period imposed by EFPIA/ABPI and be specific about which years from your sample fell outside of it during the period of data collection. I suspect that this has implications for the growing number of disclosing companies you report in your study and the increasing value / number of payments. This should also be listed as a limitation towards the end of the paper. P8 line 11 - please support the statement on “virtually all companies” with a reference. P8 line 14 - the statement about payments being disclosed or not in Disclosure UK is problematic. To establish whether payments to POs were disclosed in Disclosure UK you’d need to examine the recipients of each payment listed as a HCO in Disclosure UK for each company (online checks) and then make this determination. Can you confirm this is what you did (this was not mentioned in the supplementary information)? This recent paper quantified the extent of overlap between disclosures on websites and in Disclosure UK, but this was based on forensic research https://bmjopen.bmj.com/content/13/3/e061591.full P8 please provide more detail on the integration of data from Disclosure UK and drug company websites, including issues relating to name standardisation, and, fundamentally, potential duplicate payments. P9 line 1 - there is a repeat word P11 line 8 - I would recommend being more careful when it comes to stating that H1 was confined. This is for two principal reasons (1) the number of data points (years) is quite small in your sample; (2) there can be quite substantial yearly variations related to occasional large single payments to a single patient organisation for a clinical trial, including the case of one payment Myeloma UK - incidentally a RD PO - reported in our previous BMJ Open study, which elevated payments reported for 2016 (https://www.bmj.com/content/365/bmj.l1806 - see tables with top 10 payments and yearly sums). I think you could strengthen your argument considerably if you considered at least the top 10 payments for each year to get a bit more context to substantiate your claim. In other words, getting some insight into the nature of the payments would be important to substantiating the claim about the overall growth of payment value. P 11 line 18 - I think and should be replaced with or P16 line 19-20 - the conflicts of interests exist objectively whether there is transparency or not. What is increased by the lack of transparency is COIS being undetected and/or not managed appropriately. Line 20 and below - the claim about the dominance of certain funders needs to be phrased more cautiously. This is in the light of
--	--

	our paper on the reporting of payments to POs in the UK (https://bmjopen.bmj.com/content/10/9/e037351), which is consistent with research in research from other countries. Specifically, it may well be that these patient organisations' annual reports submitted to the Charity Commission reveal additional funders and/or payments. You make a related point in the limitations but it should also be introduced here as it has implications for the interpretation of your findings.
--	--

VERSION 2 – AUTHOR RESPONSE

#	Reviewer 1 comments	Replies (please note that pages and lines refer to the clean version of the manuscript)
	General comments	
G1	However, I still think the authors do not properly and sufficiently address the issue of missing data. The authors say in the Methods: "If payments were not disclosed in Disclosure UK nor in the company's website, we assumed the company was (sic) did not make any payments to patient organisations in a given year which is commonly assumed in the literature." However, in their rebuttal letter they quote from the UK study referenced in this sentence as following: "A missing report may indicate no payments or a failure to disclose". Their statement in the paper is thus incomplete. Indeed, the literature is very clear on the point that lack of payment reports from companies is more likely to represent a failure to disclose, or a removal of reports from the public domain. What strongly indicates that they are dealing with the latter (i.e., missing data) is not only the increase in the number of companies reporting but also (1) that there are apparently no companies that reported in 2020 but not in 2018 or 2019, and (2) the number of companies reporting in 2018 and 2019 are fewer than what was reported in the previous UK study (as well as in the much smaller Nordic countries).	Thank you for your comment. We have decided to follow your suggestion, as well as address a comment from Reviewer 2, by shifting our analytical focus to solely look at data from 2020 from companies' websites and the Disclosure UK healthcare organisations (HCOs) database. Therefore, comparisons across years were not part of our study. Although we acknowledge that some companies may not have reported their payments in 2020, resulting in some missing data, we have made the assumption - commonly done in the literature - that if we could not find a disclosure from a company after screening both data sources, said company made no payments to UK patient organizations in 2020. We believe this assumption is reasonable, especially given the ABPI requirement to maintain payment information publicly available for at least 3 years and the data collection taking place in early 2022. Finally, we discuss missing data and reporting issues in the discussion (p. 17, l. 12-25).

G2	In addition, the authors chose not to comment on the fact that the previous UK study, which included data from 2012-2016, found reported payments in 2015 and 2016 that were larger in total than those reported for 2018 in this study; and the value in 2016 was almost £21 million, compared to the £18 million in 2020 reported by the authors. Instead, they refer to the average across all years, 2012-2016, in the previous UK study which “erases” the higher values that might complicate their argument. Overall, the claim (hypothesis 1) that the present “results confirm our expectations of increasing industry funding as expressed in Hypothesis 1” seems very shaky, therefore, as payments reported in 2016 were larger than what is reported here for 2018-2020. The authors main counter-argument is most likely that there was an increase in payments among the companies that disclose consistently between 2018-2020. However, without having a longer time series it is difficult to know if this reflects “random” fluctuations or is a “real” effect. I suggest the authors delete this analysis and hypothesis 1 from their study.	Thank you for your comment. As per your suggestion and to address a comment from Reviewer 2, we have decided to shift our analytical focus and solely look at data from 2020 from companies' websites and the Disclosure UK healthcare organisations (HCOs) database. As such, comparisons across years were not part of our study. Therefore, we have removed Hypothesis 1 from our analysis (p. 5, l. 32-35).
G3	Yet, the fact that they seem to have “complete” data from 37 companies immediately raises the question what happens if the authors only include those companies in their analysis? The authors make the following claim in their limitation section: “While this might bias our results, the impact of this was considered to be limited. Most notably, despite the differences in sample size, absolute values of payments are very similar when considering only companies that consistently disclosed payments across years (n=37). For example, in 2020, payments from those companies that disclosed consistently across the study period amounted to £15.5 million versus £18 million when any payment disclosed in that year is considered (86%).” The statement that the “impact of this was considered to be limited” is not really justified but can be tested by running the analyses with the 37 companies instead. This should not be very problematic since the value of payments from these firms comprise 93% and 86% of the total value of payments for 2019 and 2020, respectively.	Thank you for this helpful suggestion! We have decided to follow your suggestion, as well as address a comment from Reviewer 2, by shifting our analytical focus to solely look at data from 2020 from companies' websites and the Disclosure UK healthcare organizations (HCOs) database (p. 7, l. 3-21). This, as noted in your comment, addresses the potential issues due to differences in sample sizes across years.

	I suggest the authors run their analyses (especially statistical tests) with the 37 companies instead. Alternatively, they could run the analysis with 2020 data only.	
#	Reviewer 2 comments	Replies
General comments		
G1	The fact that companies disclosing consistently had similar payment patterns does not allow you to make inferences about non-disclosing companies. The pattern and composition of payments can be determined by many variables which you are not controlling for. To use an example from a related field, company size and focus on narrow drug portfolios seem to be strongly associated with a high share of R&D payments.	Thank you for your comment! As per a comment by Reviewer 1, we have amended our analysis and now only consider payments in 2020 (p. 7, l. 3-4. As such, we have included all payments from companies' websites and the Disclosure UK Healthcare organisation database for this year (p. 7, l. 3-21). While we appreciate that there are missing data for 2020, the bias you are referring to, which mostly arose due to cross-year comparisons, would be less significant here. We discuss missing data and reporting issues in the discussion (p. 17, l. 12-25).
Non-general comments		
1	P5 line 2 The first sentence could be made more precise (e.g. some patient organisations also represent and support the needs of carers) and supported by references. Main points from this sentence are also repeated below in para 1 and 2, which makes at least some of it redundant.	Thank you for your comment. We have now amended the first two sentences and the third paragraph to avoid repetitions (p. 4, l. 2-4). “Patient organisations – not-for-profit organisations mainly composed of patients and/or caregivers that represent and support the needs of patients or caregivers ^{1 2} – play an important role in the development, regulatory review, and adoption of new drugs.”
2	P5 line 3 In the sentence with the definition of patient organisations please be specific about its source. If it is an industry definition, this must be made clear to the reader.	Thank you for your comment. While we agree that it is important to clearly state the definition of patient organisations used in the analysis, we feel that we do so by carefully explaining our rationale in the Methods section

		(p. 8, lines 3-13). Also, we believe that not explicitly saying that we used EFPIA's definition in the first paragraph of our study does not reduce transparency.
3	P5 line 33 – a definition of rare diseases would be helpful as they vary across jurisdictions	Thank you for raising this. We have now added a sentence indicating the prevalence point up to which diseases are considered rare in the relevant jurisdiction and provided a reference (p. 4, l. 32-33). “In the UK, diseases are defined rare if they affect up to 5 people in 10,000.”
4	P5 line 35 – what do you mean by “fragmented”?	Many thanks for your comment. By fragmented, we meant to describe the low prevalence, high number and diversity of diseases that qualify as rare. We have now rephrased the original sentence as follows: “The low prevalence of rare diseases and their different aetiology, coupled with the lack of interest from policymakers and manufacturers, who often prioritise more profitable and prevalent diseases, has necessitated the formation of patient organisations to advocate for the needs of rare disease patients” (p. 4, l. 33-34).
5	P5 line 22-3 – for further context, a recently published study has considered concentration of payments in UK countries looking jointly at HCOs and POs https://bmjopen.bmj.com/content/13/3/e061591.full	Many thanks for your suggestion, we have referenced the paper you mention in the introduction (p. 4, l. 16-20).
6	P6 H3 – this hypothesis is not entirely convincing given the already well-documented process of “orphanisation” of drug development and the fact that many companies use the drug discovery model established in the field of rare diseases in other fields. Given this, companies may, in fact, have more incentives to support RD POs over non-RD POs. You may agree or disagree with this suggestion but it is important to foreground the	Thank you for your comment. We have revised the hypothesis regarding the difference in funding of rare and non-rare-focused patient organisations as follows “Furthermore, we hypothesise that patient

	hypothesis appropriately in the existing research (i.e. the processes of orphanisation, salami-slicing of diseases etc).	organisations targeting rare diseases would receive less overall funding due to their low prevalence. However, the existing incentives, high costs and consequent profitability of some orphan-designated drugs might affect the proportion of funding directed towards these organisations” (p. 5, l. 36-39). Additionally, we have added the following paragraph in the discussion where we contextualise our findings in light of the high return of investments of some rare disease medicines (p. 16, l. 4-7): “Furthermore, rare diseases have proved a lucrative asset for pharmaceutical companies.⁴³ The additional market protection granted to orphan-designated product and the often higher willingness to pay from payers has led companies to increasingly focus on these medicines, which can offer a high return on investment.^{27 28”}
7	P8 para 1 – it would be important to inform the readers about the 3-year mandatory data retention period imposed by EFPIA/ABPI and be specific about which years from your sample fell outside of it during the period of data collection. I suspect that this has implications for the growing number of disclosing companies you report in your study and the increasing value / number of payments. This should also be listed as a limitation towards the end of the paper.	Thanks for raising this point. We have now made it clear that companies are mandated to publish payments to patient organisations only for three years following such payment(s) (p. 7, l. 6-8). We also explained the impact that this might have had on our sample size in the discussion and limitation sections (p. 17, l. 12-14).
8	P8 line 11 – please support the statement on “virtually all companies” with a reference.	Thank you for your comment. We have amended the text to reflect that the PMCPA affects both ABPI members and non-members operating in the UK, and ultimately decided to remove the statement that “virtually all companies” abide by the PMCPA code (p. 6, l. 8-10).

9	P8 line 14 - the statement about payments being disclosed or not in Disclosure UK is problematic. To establish whether payments to POs were disclosed in Disclosure UK you'd need to examine the recipients of each payment listed as a HCO in Disclosure UK for each company (online checks) and then make this determination. Can you confirm this is what you did (this was not mentioned in the supplementary information)? This recent paper quantified the extent of overlap between disclosures on websites and in Disclosure UK, but this was based on forensic research https://bmjopen.bmj.com/content/13/3/e061591.full	Thank you for your comment. Initially, we did not include payments reported in the Disclosure UK healthcare organisations (HCOs) database in our analysis as we were not aware of the extent of the misreporting by pharmaceutical companies. However, in response to Reviewer 1's comment to restrict our analysis to 2020 and in light of your recently co-authored paper, we decided to incorporate the Disclosure UK HCOs database along with information available on companies' websites to ensure the most accurate snapshot of payments to patient organisations. We have updated the methodology sections in both the main document (p. 7, l. 3-24) and the Supplemental Materials (p. 1, l. 4-18) accordingly.
10	P8 please provide more detail on the integration of data from Disclosure UK and drug company websites, including issues relating to name standardisation, and, fundamentally, potential duplicate payments.	Thank you for your comment. We have now updated the methodology sections in both the main document (p. 7, l. 3-24) and the Supplemental Materials (p. 1, l. 4-18) to reflect the changes in the data collection. In terms of the issues that you raise, we created a table in the Supplemental Materials (p. 4) detailing our process for standardizing patient organisation names and checked for duplicate payments between the two databases using STATA. We did not find any instances of duplicate payments in our database, as mentioned in the main document (p. 7, l. 29-30).
11	P9 line 1 - there is a repeat word	Many thanks, we have now removed the repeated word!
12	P11 line 8 - I would recommend being more careful when it comes to stating that H1 was confined. This is for two principal reasons (1) the number of data points (years) is quite small in your sample; (2) there can be quite substantial yearly variations related to occasional large single payments to a single patient organisation for a	Thank you for your comment. We have revised our methodology and will now focus on a single year (2020) instead of three years. Therefore, we have

	clinical trial, including the case of one payment Myeloma UK - incidentally a RD PO - reported in our previous BMJ Open study, which elevated payments reported for 2016 (https://www.bmj.com/content/365/bmj.l1806 - see tables with top 10 payments and yearly sums). I think you could strengthen your argument considerably if you considered at least the top 10 payments for each year to get a bit more context to substantiate your claim. In other words, getting some insight into the nature of the payments would be important to substantiating the claim about the overall growth of payment value.	removed Hypothesis 1 from our study. By focusing on a single year, any large payments made during that year will not necessarily bias our interpretation of the 2020 snapshot we aim to present.
13	P 11 line 18 - I think and should be replaced with or	Thanks, we have changed the wording!
14	P16 line 19-20 - the conflicts of interests exist objectively whether there is transparency or not. What is increased by the lack of transparency is COIS being undetected and/or not managed appropriately.	Many thanks for your comment. We have clarified this point by saying that the lack of transparency increases the risk of conflicts of interest not being properly detected and managed (p. 15, l. 21-22).
15	Line 20 and below - the claim about the dominance of certain funders needs to be phrased more cautiously. This is in the light of our paper on the reporting of payments to POs in the UK (https://bmjopen.bmj.com/content/10/9/e037351), which is consistent with research in research from other countries. Specifically, it may well be that these patient organisations' annual reports submitted to the Charity Commission reveal additional funders and/or payments. You make a related point in the limitations but it should also be introduced here as it has implications for the interpretation of your findings.	Following the suggestion of Reviewer 1 in the previous round of reviews, we have included a paragraph that provides context to our recommendations regarding the importance of financial independence of patient organisations. While we acknowledge the discrepancies in reporting practices between funders and recipients, we believe that it is still relevant to mention Abbvie's sponsorship of patient organisations that may oppose the switch to biosimilars, as this can raise concerns about potential conflicts of interest. Therefore, we have rephrased the sentence as follows (p. 16-17, l. 36-6): “Furthermore, the financial independence of patient organisations is fundamental to ensure that patients' interests are at the forefront of the organisations' agenda.⁵² Compromising this independence can have a detrimental effect and distort public health priorities. For

		example, AbbVie-sponsored patient organisations were found to strongly oppose switching to biosimilars for Humira, the company's blockbuster drug, in various countries.¹⁵ Similarly, a recent investigation uncovered strong financial connections between Novo Nordisk and UK-based patient organisations that supported the approval of the company's latest obesity drug. This, alongside other ongoing investigations, culminated in the suspension of the company from ABPI.⁵³ The strong financial ties between Novo Nordisk and patient organisations, contributing to the NICE appraisal of the company's drug, raises serious concerns about these groups' independence and might ultimately harm patients."
--	--	--

VERSION 3 – REVIEW

REVIEWER	Mulinari, Shai Lunds Universitet, Sociology My partner is employed by ICON, a global Contract Research Organization whose customers include many pharmaceutical companies
REVIEW RETURNED	18-May-2023

GENERAL COMMENTS	Thank you for addressing my concerns. I very much appreciate the work and believe the paper has improved significantly.
---

REVIEWER	Ozieranski, Piotr University of Bath
REVIEW RETURNED	30-May-2023

GENERAL COMMENTS	Thank you for addressing my earlier comments. The paper is much clearer more streamlined and will make a distinct contribution to an important field of study. I enjoyed reading it - congratulations! I have one minor outstanding comment: I'm not clear why you are adjusting payments for inflation to 2020 GBP if the disclosure reports also come from 2020. This comment refers to both the methodology and the notes under the tables.
--

VERSION 3 – AUTHOR RESPONSE

#	Reviewer 1 comments	Replies (please note that pages and lines refer to the clean version of the manuscript)
	N/A	
#	Reviewer 2 comments	Replies
1	I have one minor outstanding comment: I'm not clear why you are adjusting payments for inflation to 2020 GBP if the disclosure reports also come from 2020. This comment refers to both the methodology and the notes under the tables.	Thank you for your comment! This was an oversight due to the methodological changes part of the latest revision. We have now changed the text in the methodology (p. 7, lines 31-35; " All payments are reported in 2020 GBP ") and in the notes under the tables (p. 11, line 3, " The Supplemental Materials detail the conversion rates used, which were retrieved from the Office of National Statistics (ONS) website ").